# Hiding in the Phase: A Provably Robust Watermark for Diffusion Models

## Abstract

Recent advances in diffusion models have necessitated robust methods for image watermarking. While state-of-the-art semantic watermarks offer impressive robustness, they share a fundamental, unaddressed vulnerability: reliance on uniformly applied structural constraints. Such architectural uniformity creates a predictable attack surface, enabling mechanism-targeted attacks that exploit the embedding rules rather than the secret key. To address this critical vulnerability, we introduce Phase-Quantization Invisible Marking (PQIM), a training-free semantic framework that combines structural heterogeneity with provable robustness. PQIM's strength is twofold. First, PQIM uses a cryptographic key to generate a sparse, pseudo-random embedding subspace, eliminating global, predictable structures and encodes the watermark via distributed phase-only modulation. Second, we prove that the bit error rate decreases exponentially with redundancy (Theorem 1), providing an information-theoretic guarantee of robustness. Extensive experiments show that PQIM achieves strong robustness and competitive performance compared to existing methods against a wide array of attacks, while maintaining a favourable fidelity-robustness trade-off compared to existing methods.

## 1 Introduction

The advent of generative diffusion models Nichol & Dhariwal (2020); Song et al. (2020); Rombach et al. (2022) has transformed digital content creation, enabling large-scale production of high-fidelity synthetic media. While fostering creativity, these models also raise risks of disinformation and fraud, exemplified by the 2023 AI-generated Pentagon explosion image that briefly impacted financial markets Bond (2023). In response, governments are mandating safeguards: the U.S. now defines watermarking as essential for authenticity Biden (2023), and China began enforcing regulations recently that require visible labels and technical identifiers such as metadata or watermarks Chi (2025). This global momentum underscores the urgent need for robust content provenance solutions.

In response to these regulatory demands, digital watermarking Bender et al. (1996); Cox et al. (1997); van Schyndel et al. (1994) has emerged as a key tool, embedding invisible information into generated content for authentication and tracking. Approaches can be grouped into three categories. Post-processing methods Al-Haj (2007); Chen & Wornell (2001); Barni et al. (1998); Kundur & Hatzinakos (1997); Tsai et al. (2000) apply watermarks to final images but often harm quality and are vulnerable to removal. Fine-tuning-based methods Feng et al. (2024); Ci et al. (2024a); Fernandez et al. (2023) embed watermarks into model parameters, offering robustness but requiring costly retraining. Recent work has converged on semantic watermarking Ci et al. (2024b); Wen et al. (2023); Gunn et al. (2025); Arabi et al. (2025); Yang et al. (2024); Li et al. (2025); Huang et al. (2024) as the state-of-the-art. By embedding information in the initial noise $z_T$, it balances two core but conflicting objectives: imperceptibility, preserving output quality, and robustness, ensuring persistence under distortions and attacks.

While progress has been made, current semantic watermarking methods share a critical vulnerability: they rely on uniformly applied structural rules. For example, some impose predictable frequency patterns Wen et al. (2023), while others enforce global quantization rules Yang et al. (2024). This uniformity creates a predictable attack surface, allowing adversaries to disrupt embeddings without breaking the secret key Zhao et al. (2024).

Beyond this architectural weakness, prior works largely depend on heuristic tuning rather than a principled theory. The trade-off between fidelity and robustness is handled empirically, with no predictive framework to guide domain choice or redundancy levels Ci et al. (2024b); Wen et al. (2023); Zhang et al. (2024); Huang et al. (2024). This lack of theoretical grounding forces trial-and-error design, preventing guarantees of performance.

We propose Phase-Quantization Invisible Marking (PQIM), a semantic watermarking framework that unites high-fidelity embedding with the rigor of Quantization Index Modulation (QIM) Chen & Wornell (2001). Unlike uniform schemes, PQIM operates in a key-defined *secret subspace* of sparse, pseudo-random frequency coordinates, blocking mechanism-targeted attacks. Within this subspace, our *perceptual-structural framework* modulates only the phase spectrum of the initial noise latent, preserving amplitude statistics and ensuring $z'$ remains in the model's noise manifold. Without retraining, PQIM delivers provable robustness: we show the watermark's Bit Error Rate (BER) decreases exponentially with redundancy (Theorem 1). Together, these contributions establish PQIM as both theoretically grounded and practically resilient.

Our contributions are threefold:

- We provide the theoretically grounded error bounds for semantic watermarking framework. Theorem 1 shows that BER decreases exponentially with redundancy, enabling predictive design of reliable watermarks.

- We introduce a key-dependent embedding paradigm that uses sparse, pseudo-random frequency locations. This design removes the vulnerability of uniform, predictable patterns and resists mechanism-targeted attacks.

- We resolve the fidelity-robustness trade-off with a principled framework: phase-only modulation preserves energy distribution, while mid-band embedding ensures resilience with high image quality.

## 2 RELATED WORK

### 2.1 DENOISING DIFFUSION MODELS

Denoising Diffusion Probabilistic Models (DDPMs) Nichol & Dhariwal (2020); Song et al. (2020) synthesize data by reversing a noise-injection process. The forward process gradually adds Gaussian noise to an image latent $z_0$ over $T$ timesteps according to a fixed variance schedule $\{\beta_t\}_{t=1}^T$:

$$q(z_t|z_{t-1}) = \mathcal{N}(z_t; \sqrt{1 - \beta_t}z_{t-1}, \beta_t I) \tag{1}$$

A model $\epsilon_\theta$ is then trained to predict and remove this noise at each step. The model is typically trained by optimizing a simplified objective:

$$\mathcal{L}_\theta := \mathbb{E}_{t,z_0,\epsilon} \left[ \|\epsilon - \epsilon_\theta(\sqrt{\bar{\alpha}_t}z_0 + \sqrt{1 - \bar{\alpha}_t}\epsilon, t)\|_2^2 \right] \tag{2}$$

where $\epsilon$ is the ground truth noise. Image generation begins with a pure Gaussian noise latent, $z_T \sim \mathcal{N}(0, I)$, and iteratively applies the denoising network to produce a clean image latent $z_0$.

Denoising Diffusion Implicit Models (DDIM) Song et al. (2020) introduce a deterministic sampling process, establishing a crucial, invertible mapping between the initial noise latent $z_T$ and the final image latent $z_0$. This relationship enables $z_T$ to function as a high-dimensional semantic latent that dictates the entire generative outcome. The ability to recover $z_T$ from $z_0$ via DDIM inversion provides a direct channel for embedding and extracting hidden information.

### 2.2 IMAGE WATERMARKING

The evolution of digital watermarking—from fragile spatial methods Bender et al. (1996) to robust frequency-domain techniques Cox et al. (1997); Barni et al. (1998); Xia et al. (1998) and now to semantic approaches within generative models—is defined by a persistent trade-off where the structured patterns required for robustness themselves create detectable vulnerabilities.

**Frequency Domain Manipulation.** The conceptual legacy of frequency-based methods now extends to the latent space in the generative era. Techniques such as Tree-Ring Wen et al. (2023) exemplify this by imprinting a predefined pattern onto the Fourier transform of the initial noise latent. This fundamentally alters the latent's power spectral density, imposing a deterministic, non-Gaussian structure upon its Fourier coefficients.

**Spatial/Distribution Domain Manipulation.** A parallel class of semantic watermarks operates on the latent vector's statistical properties, systematically violating the i.i.d. Gaussian assumption. For instance, Gaussian Shading Yang et al. (2024) constrains latent values to replicated probability intervals, breaking their independence and introducing higher-order statistical dependencies. More directly, GaussMarker Li et al. (2025) partitions the latent tensor to manipulate the local mean of each block, violating the zero-mean property of standard noise. The PRC Watermark Gunn et al. (2025) imposes a deterministic, pseudorandom structure on the signs of latent values via an error-correcting code. This method preserves first-order statistics but breaks the statistical independence of the signs, embedding a complex, high-order dependency as the watermark.

**Model Parameter Manipulation.** A third paradigm modifies the generative model itself. Stable Signature Fernandez et al. (2023), for example, fine-tunes model weights to embed an invisible pattern as a statistical fingerprint across outputs. Despite differences in frequency, spatial, or weight domains, all approaches rely on structured regularity—leaving a detectable vulnerability that echoes classical watermarking challenges.

## 2.3 DISTORTIONS AND ATTACKS ON WATERMARKED IMAGES

The efficacy of any watermarking scheme is ultimately determined by its robustness against a diverse spectrum of attacks. These threats range from conventional signal-processing distortions to sophisticated, model-based attacks that leverage the power of deep generative models. Understanding this adversarial landscape is crucial for assessing the practical viability of modern watermarking techniques.

**Conventional Signal-Processing Attacks.** A watermark's resilience is traditionally evaluated against signal-processing attacks, such as transformations in the spatial or frequency domains. These methods are fundamentally constrained by a critical trade-off. Aggressive parameterization sufficient to remove a robust watermark typically incurs a perceptible, and often unacceptable, degradation of image quality.

**Advanced Generative Attacks.** Deep generative models enable a potent new class of attacks, fundamentally shifting the threat landscape. The most formidable is the regeneration attack Zhang et al. (2024), a destructive-constructive process. First, significant noise is added to a watermarked image to disrupt the signal. Second, a powerful pre-trained generative model, such as a Variational Autoencoder (VAE) van den Oord et al. (2017); Ballé et al. (2018); Cheng et al. (2020) or, more effectively, a diffusion model Nichol & Dhariwal (2020), denoises and reconstructs the image.

The attack's efficacy stems from the constructive step, where re-synthesis forces the image onto the attacker's learned data manifold. This process overwrites the original generative trace, obliterating subtle perturbations and, as proven by Zhang et al. (2024), rendering any small $l_2$-distance watermark fundamentally vulnerable. The threat paradigm thus shifts from signal resilience to an existential one against any imperceptible signal, necessitating more deeply embedded, content-linked watermarks.

## 3 METHOD

In this section, we first give a high-level overview of PQIM (Fig. 1), then formalize the embedding (Sec. 3.1), extraction and decoding (Sec. 3.2) and extension to perceptual tuning (Sec. 3.3)

To address the vulnerabilities of prior methods, we propose PQIM, the framework to exploit the *phase spectrum* of diffusion model noise for watermarking. Phase governs structural and textural information, whereas amplitude defines energy distribution. By preserving the amplitude spectrum,

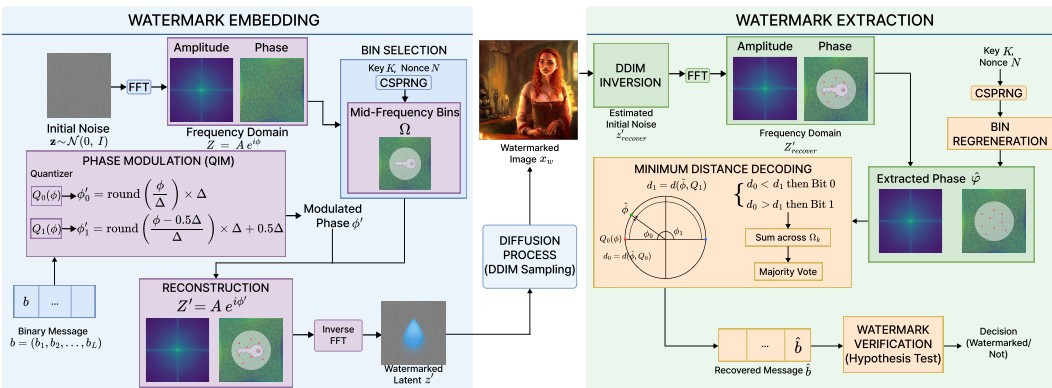

Figure 1: The PQIM framework. (Left) The embedding pipeline transforms the initial noise $z$ into a watermarked latent $z'$ by modulating its phase spectrum. (Right) The extraction pipeline recovers the message from a watermarked image by inverting the generation process and applying a minimum distance decoder to the recovered phase values.

PQIM maintains the power spectral density and second-order statistics, ensuring fidelity while embedding information in phase. This novel design keeps the modified latent $z'$ within the model's noise manifold, preventing out-of-distribution artifacts and preserving semantic coherence. Combined with Quantization Index Modulation (QIM) Chen & Wornell (2001); Sullivan et al. (2004) and a theoretically grounded robustness framework, PQIM provides a watermark that is imperceptible, provably robust, and seamlessly integrated without retraining. The overall in Fig. 1 consists of two stages: embedding and extraction.

### 3.1 WATERMARK EMBEDDING

The embedding stage modulates the i.i.d. Gaussian noise latent, $z \sim \mathcal{N}(0, I)$, to encode a binary message $b = (b_1, b_2, ..., b_L)$. The result is a watermarked latent $z'$, which initializes the diffusion process. In the following, we first describe how PQIM selects a secret subset of mid-frequency coefficients and then how it encodes each message bit by phase quantization and distributed redundancy.

**Frequency-Domain Representation and Bin Selection.** We first transform each channel $c$ of the initial noise latent $z$ with a 2D FFT and write

$$Z_c(u, v) = A_c(u, v)\, e^{i\phi_c(u,v)},$$

where $A_c(u, v) \geq 0$ and $\phi_c(u, v) \in [-\pi, \pi]$ denote the amplitude and phase of coefficient $(u, v)$. We never modify $A_c(u, v)$. All watermarking takes place by quantizing $\phi_c(u, v)$. The choice of frequency band is critical: low frequencies are robust but visually sensitive, whereas high frequencies are fragile under common distortions. Our ablations (Sec. 4.2) show that a mid-frequency band yields the fidelity–robustness trade-off, so we restrict embedding to a normalized frequency range $[0.1, 0.7]$.

Within this mid-frequency band, we first form a pool of candidate coordinates and then generate frequency bins as secret subspaces using a CSPRNG (Cryptographically Secure Pseudorandom Number Generator) initialized with a secret Key ($K$) and a unique per-image Nonce ($N$). This key-dependent sampling ensures that each user obtains a unique watermark pattern, strengthening security against targeted attacks. The selected bins are then partitioned into $L$ disjoint subsets $\Omega_1, ..., \Omega_L$, each assigned to embed a single message bit $b_k$. This distributed and secret allocation underpins the robustness and security of our embedding method.

**Phase Modulation and Robustness via Distributed Encoding.** The multi-step denoising process of a diffusion model can be regarded as an effective communication channel that introduces perturbations. We treat these perturbations as channel noise, an approach that aligns with classical information-theoretic analyses that also frame watermarking as a communication problem Moulin & O'Sullivan (2003). We show that our distributed encoding strategy, where each bit $b_k$ is encoded across all $N_{\text{bins}}$ locations in its corresponding bin set $\Omega_k$, enhances robustness significantly.

**Theorem 1** (Information-Theoretic Robustness of Distributed Encoding). *Let the decoding decision for each of the $N_{bins}$ coefficients be an i.i.d. Bernoulli random variable with an error probability of $p_e < 0.5$. For a final bit decoded by a majority vote, the bit error rate $P_B$ is upper-bounded by:*

$$P_B \leq \exp\left(-2N_{bins}\left(\tfrac{1}{2} - p_e\right)^2\right).$$

*Proof.* Let $X_k$ be a Bernoulli random variable representing the outcome of the local decision for the $k$-th coefficient, where $X_k = 1$ for a correct decision. Assuming the coefficients are sufficiently separated in frequency, the $X_k$ are i.i.d. with $P(X_k = 0) = p_e$.

While perfect independence is not guaranteed, the pseudo-random sampling from a large candidate pool makes this a reasonable working assumption for the analysis. The final bit is decoded incorrectly if the sum of correct decisions $S_N = \sum_{k=1}^{N_{bins}} X_k$ is less than the majority threshold, i.e., $S_N < N_{\text{bins}}/2$. Thus, the final bit error rate is the tail probability

$$P_B = P\left(S_N < \frac{N_{\text{bins}}}{2}\right).$$

Applying Hoeffding's inequality Hoeffding (1963) to the sample mean $\bar{X} = S_N/N_{\text{bins}}$, we obtain

$$P_B = P\left(\bar{X} - \mathbb{E}[\bar{X}] < \tfrac{1}{2} - (1 - p_e)\right) \leq \exp\left(-2N_{\text{bins}}\left(\tfrac{1}{2} - p_e\right)^2\right).$$

This bound shows that as redundancy $N_{\text{bins}}$ increases, the overall bit error rate converges to zero exponentially, achieving an effect analogous to coding gain in classical information theory. This provides a rigorous justification for our distributed encoding strategy in overcoming the inherent noise of the generative process and subsequent attacks. □

The core embedding mechanism is Quantization Index Modulation (QIM). For each bit $b_k \in \{0, 1\}$, we define two distinct quantizer codebooks, $Q_0$ and $Q_1$:

$$Q_0(\phi) = \Delta \cdot \text{round}(\phi/\Delta) \tag{3}$$

$$Q_1(\phi) = \Delta \cdot \text{round}\left(\frac{\phi - \Delta/2}{\Delta}\right) + \frac{\Delta}{2}. \tag{4}$$

The original phase $\phi_c(u, v)$ of each bin in $\Omega_k$ is replaced by a modulated phase $\phi'_c(u, v)$ using the quantizer corresponding to the message bit $b_k$:

$$\phi'_c(u, v) = Q_{b_k}(\phi_c(u, v)), \quad \forall (c, u, v) \in \Omega_k \tag{5}$$

**Roles of Hyperparameter and Trade-offs.** PQIM exposes a small set of interpretable hyperparameters that govern the robustness-fidelity-capacity trade-off. Among these, the quantization step size $\Delta$ is the primary robustness knob. Larger $\Delta$ increases the separation between the codebooks $Q_0$ and $Q_1$, widening the decoder's decision boundary and making each local decision more tolerant to noise and distortions. The redundancy parameter $N_{\text{bins}}$ controls how many coefficients contribute to each bit. Increasing $N_{\text{bins}}$ reduces the final BER $P_B$ exponentially according to Theorem 1, at the cost of using more coefficients per bit and thus reducing effective capacity. In practice, we fix the mid-frequency band $[0.1, 0.7]$ based on the ablations in Sec. 4.2 and treat $\Delta$ and $N_{\text{bins}}$ as the main design parameters.

**Reconstruction of Watermarked Noise.** To guarantee a real-valued signal after inverse FFT, we enforce Hermitian symmetry by pairing each modulated phase with its conjugate counterpart. The reconstructed spectrum $Z'_c = A_c e^{i\phi'_c}$ combines the original amplitude $A_c$ with the modulated phase $\phi'_c$, yielding the watermarked latent $z' = \mathcal{F}^{-1}(Z'_c)$.

**Image Generation.** The watermarked latent $z'$ is used as the initial latent for the diffusion model. Standard DDIM sampling then denoises $z'$ to $z_0$, and decoding produces the final watermarked image latent $z_0^w$.

## 3.2 WATERMARK EXTRACTION

This subsection explains how we approximately recover the initial watermarked noise latent via DDIM inversion and then perform robust decoding from the noisy phase estimates.

**Noise Retrieval via DDIM Inversion.** A watermarked image $x_w$ is first encoded into the latent space, $z'_{\text{recover}} = \mathcal{E}(x_w)$. Deterministic DDIM inversion then estimates the initial noise $z'_{\text{recover}} \approx z'$. These approximations inevitably introduce errors, making a robust decoding step essential to find an initial latent.

**Minimum Distance Decoding.** To recover the hidden message, we apply FFT to $z'_{\text{recover}}$ to obtain the phase spectrum $\hat{\phi}$. Using the secret key $K$ and the nonce $N$, the bin locations $\Omega_1, ..., \Omega_L$ are regenerated. Each bit $\hat{b}_k$ is decoded by minimum distance:

$$\hat{b}_k = \arg\min_{j \in \{0,1\}} \sum_{(c,u,v) \in \Omega_k} d(\hat{\phi}_c(u,v), Q_j(\hat{\phi}_c(u,v))) \tag{6}$$

with angular distance $d(\phi_1, \phi_2) = \min(|\phi_1 - \phi_2|, 2\pi - |\phi_1 - \phi_2|)$. Summation across bins acts as majority voting, mitigating errors from inversion noise and attacks.

**Watermark Verification.** The recovered bits $\hat{b}$ are evaluated through a hypothesis test: $H_0$ assumes no watermark (bit accuracy $\approx 0.5$), while $H_1$ assumes its presence. A decision threshold $\tau$ is calibrated via ROC analysis Cox et al. (2007) on non-watermarked images to fix the False Positive Rate (FPR), set to 1% in our experiments. The watermark is verified if $(1 - \text{BER}(\hat{b}, b)) > \tau$, ensuring statistical confidence while controlling false detections.

**Threat Model.** We follow the standard semantic watermarking setting used in prior work Wen et al. (2023); Yang et al. (2024). The adversary knows the watermarking and decoding algorithms, all hyperparameters, and the underlying diffusion model, but does not know the secret key $K$, the per-image nonce $N$, or the embedded message $b$. The adversary can observe watermarked images and apply arbitrary image- or latent sapce transformations in Section 2.3. We consider two performance criteria: (1) reliable detection at a fixed false positive rate and (2) accurate message recovery, measured by bit accuracy.

## 3.3 EXTENSION TO PERCEPTUAL TUNING

Intuitively, this optimization adapts the phase perturbations to each image so as to preserve perceptual quality while maintaining decodability. One possible application of this research is to extend perceptual watermarking with a fully recoverable bit string. To achieve this, we adapt and adjust the optimization pipeline from ZoDiac Zhang et al. (2024), a state-of-the-art method used for tree-ring watermarking Wen et al. (2023), and apply it to watermarking algorithms capable of bit string embedding. Our objective is to embed a message into an original image latent $z_{\text{orig}}$ to produce a watermarked image latent $z_w$ that is visually faithful to the original while allowing for perfect message decoding. This is formulated as an optimization problem that minimizes a composite loss function, $\mathcal{L}_{\text{total}}$, which balances multiple perceptual quality factors: $z_w^* = \arg\min_{z_w} \mathcal{L}_{\text{total}}$. Here, $z_w^*$ is the final watermarked image latent. Following the ZoDiac framework, the total loss function is a weighted sum of four distinct metrics: an image-level $l_2$ distance, the Watson-VGG perceptual distance ($\mathcal{L}_{\text{Watson}}$), the Structural Similarity Index (SSIM), and an FFT-level $l_1$ distance. The full objective function is: $\mathcal{L}_{\text{total}} = \lambda_{\text{L2}} \mathcal{L}_{\text{L2}} + \lambda_{\text{Watson}} \mathcal{L}_{\text{Watson}} + \lambda_{\text{SSIM}} \mathcal{L}_{\text{SSIM}} + \lambda_{\text{FFT-WM}} \mathcal{L}_{\text{FFT-WM}}$.

## 4 EXPERIMENTS

We evaluate PQIM in two complementary operating modes: (1) a generative watermarking pipeline, where the watermark is embedded into the initial latent of a text-to-image diffusion model and (2) a post-hoc perceptual tuning pipeline, where the watermark is optimized into a given image latent. For each mode, we report quantitative results and qualitative visual comparisons, in addition to ablation studies confirming our design choices.

Table 1: Fidelity comparison (FID / CLIP) within the generative watermarking pipeline. PQIM preserves generation quality better than competing robust methods.

| Method | FID ↓ | CLIP ↑ |
|---|---|---|
| DWTDCT | 25.2665 / 25.2623 / 25.4606 | 35.6771 / 35.6771 / 33.3747 |
| DWTDCTSVD | 25.5931 / 25.5890 / 25.8153 | 36.3561 / 35.6454 / 33.3519 |
| Tree Ring | 25.5580 / 25.8852 / 26.2841 | 36.2252 / 33.5572 / 27.8984 |
| Gaussian Shading | 25.1446 / 25.2029 / **25.1868** | 35.7904 / **35.6952** / **35.7806** |
| Stable Signature | 33.1977 / - / - | 36.3235 / - / - |
| PRC | 27.9710 / 25.6650 / 27.1528 | 34.8561 / 33.9706 / 32.6206 |
| PQIM | **25.0154 / 25.0141** / 25.7876 | **36.4645** / **35.6952** / 33.7538 |

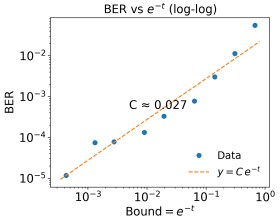

Figure 2: Empirical BER vs. the Hoeffding bound $e^{-t}$ for different redudnacy levels $N_{\text{bins}}$.

Table 2: Comparison results of TPR@1%FPR / Bit Accuracy under various image distortions.

| Methods | DM | Clean | JPEG | Blur | G. Noise | Bright | Resize | S. Noise | Average |
|---|---|---|---|---|---|---|---|---|---|
| DwtDct | | 1.0000 / 0.9999 | 0.6470 / 0.6094 | 0.0170 / 0.4973 | 1.0000 / 0.9999 | 0.9400 / 0.7910 | 1.0000 / 0.9999 | 0.1090 / 0.5397 | 0.6733 / 0.7767 |
| DwtDctSvd | | 1.0000 / 0.9760 | 1.0000 / 0.9154 | 1.0000 / 0.9112 | 1.0000 / 0.9760 | 0.0340 / 0.5091 | 1.0000 / 0.9642 | 0.9860 / 0.6768 | 0.8600 / 0.8470 |
| Tree-Ring | | 1.0000 / - | 0.9250 / - | 0.6910 / - | 1.0000 / - | 0.9930 / - | 0.9990 / - | 0.9290 / - | 0.9339 / - |
| Gaussian Shading | SD V2.1 | 1.0000 / 1.0000 | 0.9990 / 0.9703 | 1.0000 / 0.9528 | 0.9000 / 0.9528 | 0.9440 / 0.9991 | 1.0000 / 0.9996 | 0.9970 / 0.8391 | 0.9771 / **0.9591** |
| Stable Signature | | 1.0000 / 0.9934 | 0.8690 / 0.7513 | 0.0130 / 0.4168 | 0.1100 / 0.5110 | 0.9250 / 0.9364 | 0.9421 / 0.7753 | 0.1407 / 0.6160 | 0.5714 / 0.7143 |
| PRC | | 1.0000 / 1.0000 | 0.8249 / 0.8224 | 0.0349 / 0.0269 | 0.1480 / 0.1441 | 0.9542 / 0.9535 | 1.0000 / 1.0000 | 0.0150 / 0.0079 | 0.5681 / 0.5650 |
| **PQIM** | | 1.0000 / 1.0000 | 1.0000 / 0.9850 | 1.0000 / 0.9303 | 0.9070 / 0.8279 | 0.9980 / 0.9970 | 1.0000 / 0.9999 | 0.9640 / 0.8417 | **0.9813** / 0.9403 |
| DwtDct | | 1.0000 / 0.9985 | 0.7260 / 0.6167 | 0.0160 / 0.4986 | 1.0000 / 0.9999 | 0.9250 / 0.7826 | 1.0000 / 0.9998 | 0.1520 / 0.5399 | 0.6884 / 0.7766 |
| DwtDctSvd | | 1.0000 / 0.9718 | 1.0000 / 0.9126 | 1.0000 / 0.9026 | 1.0000 / 0.9718 | 0.0300 / 0.5099 | 1.0000 / 0.9586 | 0.9780 / 0.6735 | 0.8583 / 0.8430 |
| Tree-Ring | | 1.0000 / - | 0.9880 / - | 0.7520 / - | 1.0000 / - | 0.9970 / - | 1.0000 / - | 0.9610 / - | 0.9569 / - |
| Gaussian Shading | SD V2.0 | 1.0000 / 0.9999 | 0.9990 / 0.9818 | 1.0000 / 0.9498 | 0.8640 / 0.7173 | 0.9990 / 0.9721 | 1.0000 / 0.9993 | 0.9960 / 0.8306 | 0.9797 / 0.9215 |
| PRC | | 1.0000 / 1.0000 | 0.8473 / 0.8470 | 0.0419 / 0.0381 | 0.2026 / 0.1985 | 0.9611 / 0.9610 | 0.9960 / 0.9960 | 0.0320 / 0.0256 | 0.5830 / 0.5809 |
| **PQIM** | | 1.0000 / 1.0000 | 1.0000 / 0.9855 | 1.0000 / 0.9188 | 0.9790 / 0.8094 | 0.9990 / 0.9973 | 1.0000 / 0.9997 | 1.0000 / 0.8443 | **0.9969** / **0.9364** |
| DwtDct | | 0.9480 / 0.9744 | 0.7420 / 0.6077 | 0.0050 / 0.4953 | 0.9480 / 0.9743 | 0.8850 / 0.7709 | 0.9480 / 0.9743 | 0.1130 / 0.5386 | 0.6556 / 0.7622 |
| DwtDctSvd | | 0.9480 / 0.9419 | 0.9480 / 0.8848 | 0.9470 / 0.8758 | 0.9480 / 0.9421 | 0.0350 / 0.5106 | 0.9480 / 0.9271 | 0.9260 / 0.6655 | 0.8143 / 0.8211 |
| Tree-Ring | | 0.9440 / - | 0.8910 / - | 0.0270 / - | 0.9440 / - | 0.9420 / - | 0.9410 / - | 0.8880 / - | 0.7967 / - |
| Gaussian Shading | SD V1.4 | 1.0000 / 0.9999 | 1.0000 / 0.9811 | 1.0000 / 0.9637 | 0.8530 / 0.6969 | 1.0000 / 0.9973 | 1.0000 / 0.9996 | 0.9990 / 0.8316 | **0.9789** / 0.9243 |
| PRC | | 0.9491 / 0.9490 | 0.7870 / 0.7870 | 0.0390 / 0.0365 | 0.1529 / 0.1494 | 0.9112 / 0.9105 | 0.9441 / 0.9440 | 0.0239 / 0.0135 | 0.5439 / 0.5414 |
| **PQIM** | | 1.0000 / 1.0000 | 1.0000 / 0.9820 | 1.0000 / 0.9193 | 0.9250 / 0.7850 | 0.9990 / 0.9972 | 0.9520 / 0.9764 | 0.9480 / 0.8158 | 0.9749 / **0.9251** |

## 4.1 EXPERIMENTAL SETUP

We used pre-trained Stable Diffusion v2.1, v2.0, and v1.4 Rombach et al. (2022) at 512x512 resolution (latent size 4x64x64). Image generation employed DDIM sampling with 50 steps and a guidance scale of 7.5. For watermark extraction, DDIM inversion was conducted for 50 steps using an empty prompt with guidance scale 1.0 to simulate unknown prompts. To ensure fairness and reproducibility, all experiments were performed with a single, fixed, publicly disclosed watermarking key, following prior benchmarks Wen et al. (2023); Ci et al. (2024b). To evaluate generated images, we used prompts from the Gustavosta/Stable-Diffusion-Prompts dataset Gustavosta (2023), measuring semantic alignment with CLIP-based scoring Radford et al. (2021); Hessel et al. (2021) and fidelity with FID Heusel et al. (2017) on the COCO 2017 validation set Lin et al. (2015). For post-hoc watermarking, 1,000 images were sampled from the same dataset using Stable Diffusion v2.1.

## 4.2 ABLATION STUDIES AND PARAMETER ANALYSIS

To build a fair basis for comparison, we conducted ablation studies to validate our theoretical claims and identify optimal hyperparameters. All results were averaged over 1,000 images and this setting was fixed for subsequent experiments.

**Validation of distributed encoding.** We empirically validate Theorem 1 by varying the redundancy $N_{\text{bins}}$ from 1 to 20 and measuring the per-coefficient error rate $p_e$ in our generative pipeline. Plotting the BER against the Hoeffding bound $e^{-t}$ with $t = 2N_{\text{bins}}(0.5 - p_e)^2$ in Fig. 5(a), the points lie close to a straight line of slope 1 (BER $\approx Ce^{-t}$) in Fig. 2, confirming the predicted exponential decay. Detailed experimental settings are deferred to appendix.

**Optimal Frequency Band Selection.** We validated that starting from 0.1 of normalized frequency range provides the best trade-off between robustness and fidelity. Embedding a 16-bit message with $N_{\text{bins}} = 4$, we observed that the [0.1, 0.2] band achieved higher SSIM Wang et al. (2004) than [0.0, 0.1] in Fig. 5(c) in appendix, while maintaining strong robustness. As shown in Fig. 9 in appendix,

$p_e$ remains relatively uniform without attacks and lower than other regions under the attacks, so one might consider including very low frequencies (e.g., [0.0, 0.1]) to improve robustness against strong attacks like JPEG or diffusion. However, starting from 0.0 incurs a clear fidelity cost, as reflected in the lower SSIM scores and degraded image quality. Therefore, to balance robustness with visual fidelity, we selected the [0.1, 0.7] range as our candidate pool.

**Impact of Quantization Step ($\Delta$).** We confirmed $\Delta$ as the key robustness parameter. Sweeping quantization step values showed that larger $\Delta$ widens the decision boundary, reducing BER to nearly 0.00001 at $\Delta = 6$, consistent with theoretical predictions in Fig 5(c).

Table 3: Visual metrics for Perceptual Tuning. PQIM achieves the best reconstruction quality (PSNR) and superior color fidelity (Hist. Correl, Delta E). Note that for LPIPS, Delta E, and EMD, lower values indicate better performance.

| Method | PSNR ↑ | LPIPS ↓ | SSIM ↑ | MSSIM ↑ | CLIP ↑ | Hist Correl ↑ | Chi-Square ↓ | Bhattacharyya ↓ | Delta E ↓ | EMD ↓ |
|---|---|---|---|---|---|---|---|---|---|---|
| PQIM | **28.7594** | 0.1247 | 0.8776 | 0.9519 | **0.9771** | **0.8911** | **180.68** | **0.2310** | **7.346** | **0.0016** |
| Tree-Ring | 25.5303 | **0.0657** | **0.9271** | **0.9663** | 0.9752 | 0.3293 | 7288.43 | 0.6276 | 11.167 | 0.0028 |
| Gaussian Shading | 21.1166 | 0.3690 | 0.7115 | 0.8281 | 0.8352 | 0.4967 | 706.09 | 0.4848 | 16.677 | 0.0049 |

## 4.3 GENERATIVE WATERMARKING

**Quantitative Evaluation.** We first evaluate PQIM in the standard generative watermarking pipeline, where the watermark is embedded into the initial latent of a diffusion model. Tab. 1 shows CLIP and FID scores on three stable diffusion backbones. PQIM matches or surpasses existing robust watermarking methods in fidelity. CLIP scores are highest or competitive across all backbones. This confirms that phase-only modulation and mid-band embedding preserve the generative prior and semantic alignment with the text prompts.

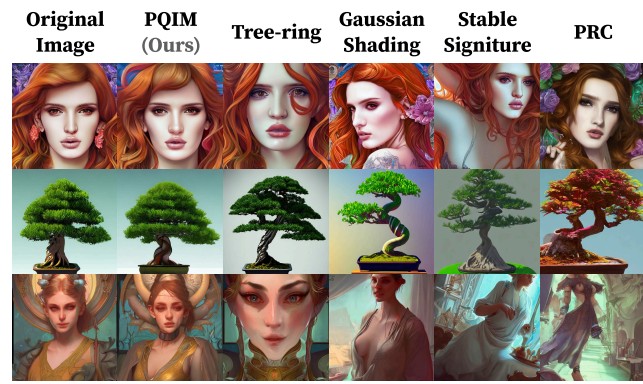

Figure 3: Generative watermarking comparison.

Tab. 2 summarizes robustness against classical distortions with TPR at 1% FPR in the generative pipeline. PQIM achieves near-perfect TPR and bit accuracy on clean images and under moderate JPEG compression and resizing. Under stronger distortions such as heavy Gaussian blur and salt-and-pepper noise, Tree-Ring and Gaussian Shading can obtain slightly higher TPR on some specific attacks, which is consistent with their design. However, PQIM avoids the performance degradation and attains the best or second-best robustness when averaged across all distortions. Overall, PQIM provides a stable robustness profile without sacrificing generation quality.

**Qualitative Evaluation** Fig. 3 presents qualitative comparisons for the generative watermarking pipeline. Across diverse prompts and styles, PQIM produces images that are less changed from original generation which means maintaining the presentation of the model.

## 4.4 POST-HOC PERCEPTUAL TUNING

**Quantitative Evaluation** We next evaluate PQIM in the post-hoc perceptual tuning pipeline from Sec. 3.3, where the watermark is optimized into the latent of a given source image. In this setting, preserving the original image appearance is paramount. Tab. 3 shows standard image-quality and color-consistency metrics. PQIM achieves the highest PSNR and Histogram Correlation, along with the lowest Delta E, indicating that it best preserves both structural details and global colorimetry. Competing methods sometimes obtain slightly higher SSIM, but often at the cost of distortions in the global color palette. PQIM's phase-oriented embedding yields a more balanced trade-off, making it arguably the most visually faithful method for post-hoc watermarking.

We also evaluate robustness of perceptually tuned images under classical distortions, Tab. 10 in the appendix. PQIM remains robust to content-preserving distortions such as JPEG compression, blur, brightness changes and resizing while showing relatively degradation under strong stochastic corruptions. Importantly, the noise levels at which PQIM loses bit accuracy yield visibly corrupted images, making such distortions less relevant for practical downstream use. In other words, PQIM explicitly trades extreme robustness under unrealistic noise for better structural and chromatic fidelity in the regime where images are still useful.

**Qualitative Evaluation**   Fig. 4 shows qualitative comparisons for the post-hoc perceptual tuning setting. A key challenge in this scenario is embedding a robust watermark while minimizing perceptual deviation from the source image. PQIM preserves fine textures and local contrast significantly better than baseline methods. These visual trends align with the quantitative results in Tab. 3. PQIM maintains high perceptual fidelity while still enabling reliable detection and message recovery under non-stochastic attacks.

**Extension to perceptual tuning.**   We further validate PQIM in a post-hoc perceptual tuning framework, comparing it against Gaussian Shading (GS) and a Tree-Ring (TR) baseline. For the optimization, we adapt ZoDiac's composite loss from Sec. 3.3 and fix the weights $(\lambda_{\text{L2}}, \lambda_{\text{Watson}}, \lambda_{\text{SSIM}}, \lambda_{\text{FFT}}) = (10.0, 0.1, 10.0, 0.1)$. After tuning all methods to achieve near-perfect bit accuracy on clean images while maintaining robustness against the attacks considered in the appendix, a clear difference in perceptual quality emerges Fig. 4 the tree-ring baseline severely alters the image's color palette, GS introduces noticeable visual artifacts, whereas our tuned PQIM uniquely preserves the original image's fidelity, maintaining both structure and color. Importantly, unlike the tree-ring baseline, which cannot carry a message, PQIM successfully embeds a recoverable bit string while maintaining this high perceptual quality.

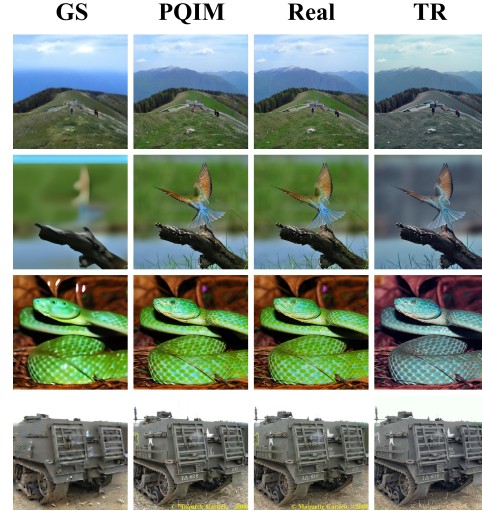

| GS | PQIM | Real | TR |

Figure 4: Perceptual tuning comparison.

## 5   CONCLUSION

In this paper, we introduced Phase-Quantization Invisible Marking (PQIM), a semantic watermarking framework that uses a key-dependent, pseudo-random phase subspace and distributed QIM embedding to avoid the uniform structural patterns of prior methods. Our analysis provides an information-theoretic bound linking redundancy and quantization to the bit error rate and our experiments show that PQIM attains high image fidelity and strong robustness across classical distortions and regeneration-based attacks in both generative and post-hoc settings, while remaining competitive in regimes where stochastic noise particularly favors existing semantic watermarks. PQIM therefore offers a practically useful and theoretically grounded complement to existing approaches for secure diffusion model watermarking.

**Limitations and Future Work.**   PQIM's limitations include finite message capacity and reliance on white-box access for extraction. Additionally, while our phase-only modulation is imperceptible under our standard operating regime, stress-test experiments at extreme payloads reveal rare failure cases in which image semantics may be distorted. We include representative visual examples in the appendix. A more systematic perceptual study of such edge cases is left for future work. Future directions include content-adaptive embedding to expand message capacity, extending the framework to other generative models and formalizing the security of the Secret Subspace against adaptive attacks. PQIM thus represents a practical step toward secure and trustworthy generative AI.

REPRODUCIBILITY STATEMENT

Our work is grounded in reproducible source codes. We utilize publicly available and officially released implementations of the pretrained models mentioned in this paper. To facilitate the precise replication of our proposed method, we have included a detailed description of our PQIM in equation and proofs. A comprehensive list of all hyperparameters and additional implementation details can be found in the Appendix. To further enhance reproducibility, the source code used for our experiments will be made available as part of the supplementary material.

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

# A APPENDIX

This appendix provides supplementary materials to support the claims made in the main paper. We include detailed implementation settings, extended ablation studies, a comprehensive analysis of robustness against classical and advanced attacks, message scalability experiments, and additional qualitative results for our perceptual tuning framework.

## A.1 IMPLEMENTATION DETAILS

All experiments were conducted utilizing the PyTorch framework on a system equipped with four NVIDIA RTX 6000 Ada Generation GPUs. For the diffusion models, we employed pre-trained checkpoints from the hugging face hub, Stable Diffusion v2.1, v2.0, and v1.4.

**Hyperparameters.** The core PQIM framework was configured with a set of hyperparameters optimized through the ablation studies detailed in Section 4. Specifically, we set the quantization step size to $\Delta = 6$ to ensure a high robustness margin while maintaining perceptual fidelity. The message was distributed across $N_{bins} = 20$ frequency coefficients per bit, embedded within the normalized frequency range of $[0.1, 0.7]$.

**Cryptographic Implementation.** To guarantee the security and reproducibility of the watermark, we implemented a cryptographic pipeline. We utilized the master secret key $K$ which is derived from a user-defined passphrase $\mathcal{P}$ using SHA-256, ensuring that high-entropy secrets are used to seed the system ($K = \text{SHA-256}(\mathcal{P})$). For selecting frequency bins, we employ the HMAC-DRBG (Hash-based Deterministic Random Bit Generator) as specified in NIST SP 800-90A Barker & Kelsey (2012). The DRBG is initialized with both the secret key $K$ and a unique, public per-image nonce $N$. This ensures that even if the same key is used across multiple images, the embedding subspaces remain computationally independent and unlinkable. Full formal proofs regarding the security properties of this implementation are provided in Section B.

**Details of Distributed Encoding Experiment.** For Fig. 2 in the main paper, we embed a 256-bit message, vary $N_{bins} \in \{1, \ldots, 20\}$ in the mid-frequency band $[0.1, 0.7]$ and average over 1,000 images. Fig. 9 complements Fig. 2 by showing the corresponding per-coefficient error rates $p_e$ and BER curves under (a) no attack, (b) regeneration attack and (c) JPEG compression.

## A.2 EXTENDED ABLATION STUDIES

To further characterize the robustness and versatility of PQIM, we conducted additional ablation studies on key components of the diffusion generation and inversion process.

**Compatibility with Various Sampling Methods.** While DPMSolver Lu et al. (2022) is a common choice for deterministic sampling, numerous other samplers exist. We tested PQIM's compatibility with other popular methods, including PNDM Liu et al. (2022) and DDIM Song et al. (2020). Tab. 4 shows that PQIM achieves near-perfect bit accuracy across all tested samplers in the absence of attacks. Furthermore, even when subjected to adversarial noise designed to disrupt the watermark, the performance remains high and consistent across samplers. This result validates that PQIM's core mechanism, which operates on the initial noise latent $z_T$, is fundamentally sampler-agnostic, allowing it to be integrated into diverse generative pipelines without modification.

Table 4: Robustness comparison (TPR@1%FPR/Bit Accuracy) under conditional distortions with different sampling methods.

| Sampler | No Distortion | JPEG | Gaussian Blur | Gaussian Noise | Brightness | Resize | SP Noise |
|---|---|---|---|---|---|---|---|
| PNDM | 1.0000 / 1.0000 | 1.0000 / 0.9823 | 1.0000 / 0.9198 | 0.9970 / 0.8179 | 0.9990 / 0.9972 | 1.0000 / 0.9998 | 0.9970 / 0.8346 |
| DDIM | 1.0000 / 1.0000 | 1.0000 / 0.9823 | 1.0000 / 0.9197 | 0.9760 / 0.8178 | 0.9990 / 0.9972 | 1.0000 / 0.9998 | 0.9970 / 0.8346 |
| DPMSolver | 1.0000 / 1.0000 | 1.0000 / 0.9850 | 1.0000 / 0.9303 | 0.9070 / 0.8219 | 0.9980 / 0.9970 | 1.0000 / 0.9999 | 0.9640 / 0.8417 |

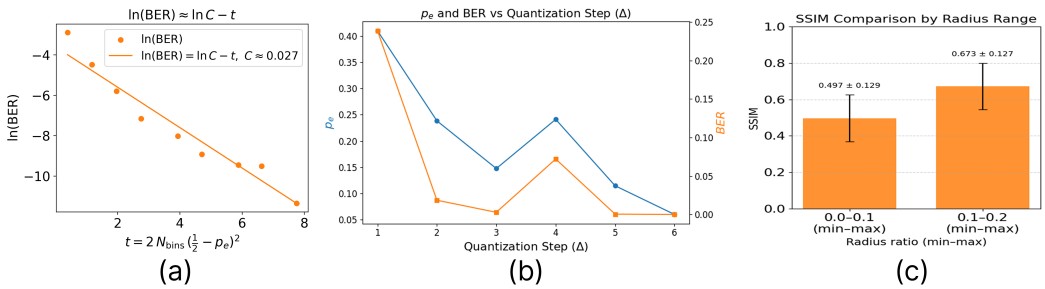

Figure 5: Experimental results of qualifying Hoeffding's inequality

**Ablation on Inference and Inversion Steps.** We further study how the number of sampling and inversion steps affects robustness. Table 5 shows TPR@1%FPR and bit accuracy for PQIM, varying the inference steps and inversion steps. Under various steps, robustness remains essentially perfect, indicating that PQIM does not rely on a fragile or finely tuned inversion schedule.

Table 5: Ablation on robustness (TPR@1%FPR / Bit Accuracy) with respect to the number of inference and inversion steps.

| Inference Step | Inversion Step | | | |
|:---:|:---:|:---:|:---:|:---:|
| | 10 | 25 | 50 | 100 |
| 10 | 1.0000 / 1.0000 | 1.0000 / 1.0000 | 1.0000 / 1.0000 | 1.0000 / 1.0000 |
| 25 | 1.0000 / 1.0000 | 1.0000 / 1.0000 | 1.0000 / 1.0000 | 1.0000 / 1.0000 |
| 50 | 1.0000 / 1.0000 | 1.0000 / 1.0000 | 1.0000 / 1.0000 | 1.0000 / 1.0000 |
| 100 | 1.0000 / 1.0000 | 1.0000 / 1.0000 | 1.0000 / 1.0000 | 1.0000 / 1.0000 |

### A.3 MESSAGE CAPACITY AND SCALABILITY

**Classical Attack Parameter Sensitivity Analysis.** To define the operational limits of PQIM, we performed extensive parameter sweeps over classical attacks, as visualized in Fig. 6. We plotted the Bit Accuracy and True Positive Rate at 1% FPR (TPR@1%FPR) against varying intensities of attacks. The curves demonstrate a stable robustness margin. The performance remains near-optimal ($> 0.99$ bit accuracy) under mild-to-moderate perturbations. Crucially, the degradation profile is monotonic and predictable, indicating that PQIM does not suffer from sudden catastrophic failures until the distortion becomes severe enough to significantly compromise the utility of the image itself.

A critical attribute of a watermarking system is its ability to scale to different message lengths without compromising performance. We evaluated PQIM's robustness across a wide range of message capacities, from 8 bits to 1024 bits. As presented in Table 6, our method sustains high TPR@1%FPR and bit accuracy under a battery of standard image distortions. While a gradual and expected degradation is observed at the largest messages due to the reduced distribution of frequency bins per bit, the performance remains exceptionally strong. This scalability confirms PQIM's suitability for diverse applications, ranging from lightweight identifiers to larger, data-rich watermarks carrying metadata or cryptographic signatures.

Table 6: Robustness under various distortions across message sizes (TPR@1%FPR/Bit Accuracy).

| message | No attack | JPEG | Gaussian Blur | Gaussian Noise | Brightness | Resize | SP Noise |
|:---:|:---:|:---:|:---:|:---:|:---:|:---:|:---:|
| 8 | 1.0000/1.0000 | 1.0000/0.9979 | 1.0000/0.9908 | 0.9670/0.9171 | 0.9980/0.9988 | 1.0000/1.0000 | 0.9860/0.9331 |
| 16 | 1.0000/1.0000 | 1.0000/0.9985 | 1.0000/0.9941 | 0.9620/0.9241 | 0.9980/0.9989 | 1.0000/1.0000 | 0.9910/0.9454 |
| 64 | 1.0000/1.0000 | 1.0000/0.9988 | 1.0000/0.9945 | 0.9600/0.9013 | 0.9980/0.9991 | 1.0000/1.0000 | 0.9940/0.9157 |
| 256 | 1.0000/1.0000 | 1.0000/0.9966 | 1.0000/0.9484 | 0.9060/0.8903 | 0.9980/0.9984 | 1.0000/0.9999 | 0.9660/0.8601 |
| 512 | 1.0000/0.9990 | 1.0000/0.9298 | 1.0000/0.8179 | 0.6970/0.7294 | 0.9980/0.9815 | 1.0000/0.9922 | 0.7500/0.7384 |
| 1024 | 1.0000/0.9397 | 1.0000/0.8417 | 1.0000/0.7396 | 0.9760/0.6552 | 0.9980/0.9065 | 1.0000/0.9165 | 1.0000/0.7095 |

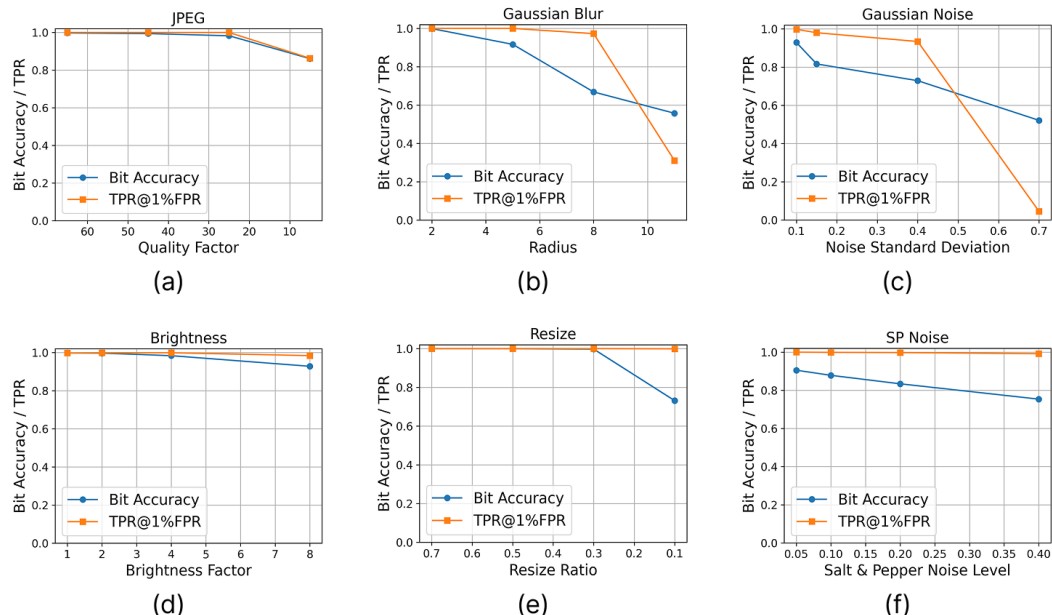

Figure 6: Parameter sensitivity analysis against classical corruptions. We report Bit Accuracy and TPR across varying attack intensities (e.g., JPEG quality, Blur radius, Noise level) to characterize the robustness margin of PQIM.

**Impact of Latent Resolution on Message Scalability.** In Tab. 6, we observed that pushing the message to 1024 bits on SD v2.1 leads to a noticeable drop in robustness. To disentangle the effect of PQIM from the limitations of the backbone, we compare SD v2.1 and SDXL at the same 1024-bit message. As summarized in Tab. 7, SDXL—which provides roughly $4\times$ higher latent spatial resolution—achieves near-perfect bit accuracy in the clean setting and significantly higher robustness under all attacks. This confirms that the degradation at large messages on SD v2.1 is primarily due to its limited latent bandwidth rather than an inherent limitation of PQIM. In principle, combining PQIM with stronger error-correcting codes and soft-decision decoding could further improve spectral efficiency within a fixed latent dimension.

Table 7: Comparison of PQIM robustness at a 1024-bit message on SD v2.1 vs SDXL (TPR@1%FPR / Bit Accuracy). SDXL's larger latent enables higher bit accuracy and robustness under all attacks, indicating that the observed limitation is due to the backbone's spatial bandwidth rather than PQIM itself.

| Attack Type | SD v2.1 (1024 bits) | SDXL (1024 bits) | Bit Acc. Gain |
|---|---|---|---|
| No Distortion | 1.0000 / 0.9397 | 1.0000 / 0.9999 | +6.02% |
| JPEG | 1.0000 / 0.8417 | 1.0000 / 0.9496 | +10.79% |
| Gaussian Blur | 1.0000 / 0.7396 | 1.0000 / 0.8485 | +10.89% |
| Gaussian Noise | 0.9760 / 0.6552 | 0.9980 / 0.7777 | +15.75% |
| Brightness | 0.9980 / 0.9065 | 1.0000 / 0.9955 | +8.90% |
| Resize | 1.0000 / 0.9165 | 1.0000 / 0.9992 | +8.27% |
| SP Noise | 1.0000 / 0.7095 | 1.0000 / 0.7310 | +2.94% |

**Extension to DiT-based diffusion models.** To verify that PQIM is not restricted to U-Net–based Stable Diffusion backbones, we also evaluate it on a DiT-based text-to-image model, PixArt-$\alpha$ Chen et al. (2023). We keep the same embedding and decoding pipeline and only replace the underlying diffusion architecture. Tab. 8 reports TPR@1%FPR and bit accuracy under the same classical distortions as in the main paper. PQIM maintains strong robustness and high bit accuracy across all attacks, indicating that our phase-only, frequency-domain embedding strategy naturally transfers to transformer-based diffusion models.

Table 8: Robustness of PQIM on a DiT-based diffusion model (PixArt-$\alpha$ Chen et al. (2023)) under various image distortions (TPR@1%FPR / Bit Accuracy).

| PixArt-$\alpha$ | No Distortion | JPEG | Gaussian Blur | Gaussian Noise | Brightness | Resize | SP Noise |
|---|---|---|---|---|---|---|---|
| **PQIM** | 1.0000 / 0.9999 | 1.0000 / 0.9642 | 0.9980 / 0.8689 | 0.9980 / 0.9236 | 0.9970 / 0.9805 | 1.0000 / 0.9947 | 0.9980 / 0.8173 |

## A.4 EVALUATION ON GENERATIVE WATERMARKING PIPELINE

In this section, we evaluate the efficacy of PQIM when integrated into the standard diffusion generation process. Our primary focus here is twofold: ensuring the watermark does not degrade the generative quality and validating its resilience against sophisticated model-based removal attempts.

**Generative Fidelity and Semantic Alignment.** For a watermark embedded during generation, it is crucial that the resulting image $x_0$ maintains high perceptual quality and semantic alignment with the text prompt. Table 1 presents the quantitative results. PQIM achieves an FID score comparable to the non-watermarked baseline. Crucially, our method maintains a high CLIP score, confirming that the phase-based modulation preserves the semantic content effectively.

**Resilience against Advanced Generative Attacks.** Watermarks embedded during generation are particularly susceptible to regeneration attacks, where adversaries use VAEs or diffusion models to reconstruct the image. Table 9 summarizes the robustness of generated images against VAE-based compression (VAE 1 Cheng et al. (2020), VAE 2 Ballé et al. (2018)) and Diffusion-based regeneration Song et al. (2020). PQIM achieves $1.0000$ TPR under all tested advanced attacks, while maintaining bit accuracy that is competitive with the strongest existing methods. PQIM avoids the large performance drops observed for certain advanced attacks and maintains reliably high detection performance across attack types. This indicates that PQIM does not introduce new weaknesses under advanced, model-based removal attempts.

Table 9: Robustness of Generative Watermarking against advanced attacks (TPR@1%FPR / Bit Accuracy). PQIM shows consistently strong robustness and competitive bit accuracy across all attacks, with notable strength under diffusion-based regeneration.

| Method | VAE 1 Cheng et al. (2020) | VAE 2 Ballé et al. (2018) | Diffusion Song et al. (2020) | Average |
|---|---|---|---|---|
| DWT-DCT | 0.0668 / 0.5235 | 0.0780 / 0.5244 | 0.0810 / 0.5309 | 0. 0753 / 0.5263 |
| DWT-DCT-SVD | **1.0000** / 0.7686 | 0.9990 / 0.7540 | 0.9707 / 0.6518 | 0.9899 / 0.7248 |
| Tree-Ring | 0.9850 / – | 0.9750 / – | 0.9370 / – | 0.9657 / - |
| Gaussian Shading | **1.0000** / **0.9536** | **1.0000** / **0.9440** | **1.0000** / 0.8847 | **1.0000** / 0.9274 |
| Stable Signature | 0.2290 / 0.5299 | 0.1496 / 0.5002 | 0.0506 / 0.4684 | 0.1431 / 0.5095 |
| PRC | 0.3470 / 0.3455 | 0.2879 / 0.2824 | 0.0489 / 0.0450 | 0.2279 / 0.2243 |
| **PQIM** | **1.0000** / 0.9374 | **1.0000** / 0.9238 | **1.0000** / 0.9233 | **1.0000** / **0.9282** |

## A.5 EVALUATION ON PERCEPTUAL TUNING FRAMEWORK

We extend PQIM to the *post-hoc* watermarking setting, utilizing an optimization-based perceptual tuning framework. In this context, maintaining the fidelity of the original image is paramount. To rigorously evaluate the visual preservation capabilities, we employed a comprehensive set of metrics focusing on both structural integrity and color consistency.

**Evaluation Metrics and Rationale.** To ensure a multifaceted assessment of image quality, we adopted the following metrics:

- **Structural Similarity Index Measure (SSIM):** To evaluate the structural fidelity of the generated images, we employed SSIM, which assesses visual similarity based on luminance, contrast, and structural information, aligning with human visual perception Wang et al. (2004). It is defined as:

$$\text{SSIM}(x,y) = \frac{(2\mu_x\mu_y + C_1)(2\sigma_{xy} + C_2)}{(\mu_x^2 + \mu_y^2 + C_1)(\sigma_x^2 + \sigma_y^2 + C_2)} \tag{7}$$

where $\mu$ and $\sigma$ represent the mean and variance of the images $x$ and $y$.

- **Histogram Correlation (Color Similarity):** To measure the similarity of global color distributions between images, we calculated the Histogram Correlation metric. This metric quantifies the statistical correlation between the color histograms of the two images Swain & Ballard (1991), computed as:

$$d(H_1, H_2) = \frac{\sum_I (H_1(I) - \bar{H}_1)(H_2(I) - \bar{H}_2)}{\sqrt{\sum_I (H_1(I) - \bar{H}_1)^2 \sum_I (H_2(I) - \bar{H}_2)^2}} \tag{8}$$

where $H_1$ and $H_2$ are the histograms of the original and watermarked images, respectively. A value closer to 1 indicates a higher correlation in color distribution.

**Quantitative Results on Visual Preservation.** Tab. 3 presents the comparative results of *post-hoc* watermarking methods. PQIM demonstrates a balance between structural preservation and color fidelity. PQIM achieves the highest PSNR and a Histogram Correlation, along with the lowest Delta E. This indicates that while competing methods may preserve edges (structure), they often distort the global color palette. PQIM's phase-oriented embedding successfully preserves both the pixel-level details and the chromatic characteristics of the source image, making it arguably the most visually faithful method for post-hoc watermarking.

**Robustness against Classical Attacks.** We subjected the perceptually tuned images to a battery of classical distortions to verify the watermark's persistence. As shown in Tab. 10, PQIM maintains robustness under most classical distortions. Regarding noise-based distortions, we observe a characteristic sensitivity. In particular, Tree-Ring and Gaussian Shading achieve higher TPR/bit accuracy under heavy Gaussian and salt-and-pepper noise, which is consistent with their design. This behavior is theoretically consistent with our design principles. Since PQIM embeds information by modulating the phase spectrum—which encodes the image's structural integrity—unstructured stochastic noise tends to disrupt these precise phase relationships more aggressively than global transformations. This represents a deliberate trade-off, prioritizing the superior structural and chromatic fidelity demonstrated in Table 3 while maintaining practical resilience against non-stochastic distortions.

Table 10: TPR@1%FPR / Bit Accuracy for Perceptually Tuned images under classical corruptions. PQIM retains strong robustness, particularly against geometric and signal processing attacks.

| Method | No distortion | Brightness | Gaussian Blur | Gaussian Noise | JPEG | Resize | SP Noise | Contrast |
|---|---|---|---|---|---|---|---|---|
| Tree | 1.0000 / – | 0.9890 / 0.9983 | 0.9940 / 0.9992 | 0.7820 / 0.9393 | 0.9730 / 0.9964 | 0.9990 / 0.9997 | 0.8730 / 0.9822 | 0.9970 / 0.9996 |
| GS | 0.9990 / 1.0000 | 0.9980 / 0.9502 | 0.9970 / 0.8742 | 0.8060 / 0.6547 | 0.9900 / 0.8530 | 0.9990 / 0.9764 | 0.9190 / 0.6806 | 0.9990 / 0.9757 |
| **PQIM** | 1.0000 / 1.0000 | 0.9980 / 0.9446 | 0.9170 / 0.6402 | 0.3270 / 0.5551 | 0.9430 / 0.7770 | 1.0000 / 0.9842 | 0.3020 / 0.5568 | 1.0000 / 0.9945 |

**Robustness against Advanced Attacks.** We evaluated the tuned images against focused regeneration attacks, which represent the most severe threat to post-hoc watermarks. Table 11 presents the results. PQIM remains detectability in the most scenario. However, regarding bit accuracy and VAE-based regeneration, we observe a performance gap compared to baselines. We attribute this to the strict perceptual constraints imposed by our tuning framework. While baselines like Tree-Ring achieve higher robustness metrics, they often do so at the cost of the significant visual and colorimetric degradation observed in Tab. 3. In contrast, PQIM prioritizes the preservation of the source image's integrity, accepting a trade-off in absolute bit accuracy while maintaining a satisfactory detection rate ($> 94\%$) and superior perceptual fidelity.

Table 11: Robustness comparison (TPR@1%FPR / Bit Accuracy) against focused diffusion-based attacks in the Perceptual Tuning setting.

| Method | Diff Attacker 60 | Cheng2020 | BMSHJ2018 |
|---|---|---|---|
| Tree-Ring | 0.9500 / 0.9937 | 0.9920 / 0.9982 | 0.9820 / 0.9972 |
| GS | 0.9980 / 0.9173 | 0.9950 / 0.8357 | 0.9990 / 0.8455 |
| **PQIM** | 0.9990 / 0.8498 | 0.9440 / 0.7618 | 0.9500 / 0.7618 |

### A.6   ADDITIONAL QUALITATIVE RESULTS.

We provide an extensive gallery of generated images to visually demonstrate the capabilities of PQIM across two distinct operating modes: the generative watermarking pipeline and the post-hoc perceptual tuning framework.

**Generative Watermarking.** Fig. 7 presents additional comparisons with other watermarking methods. PQIM maintains the underlying diffusion model's generative prior. The results confirm that modulating the phase spectrum at the initial noise stage preserves both the semantic alignment with text prompts and the high-frequency textural fidelity, rendering the watermark imperceptible across diverse subjects and styles.

Figure 7: Extended qualitative results for Generative Watermarking. The images demonstrate that embedding the watermark into the initial noise phase preserves the generative quality and semantic coherence without introducing visible artifacts.

**GS** **PQIM** **Real** **TR**

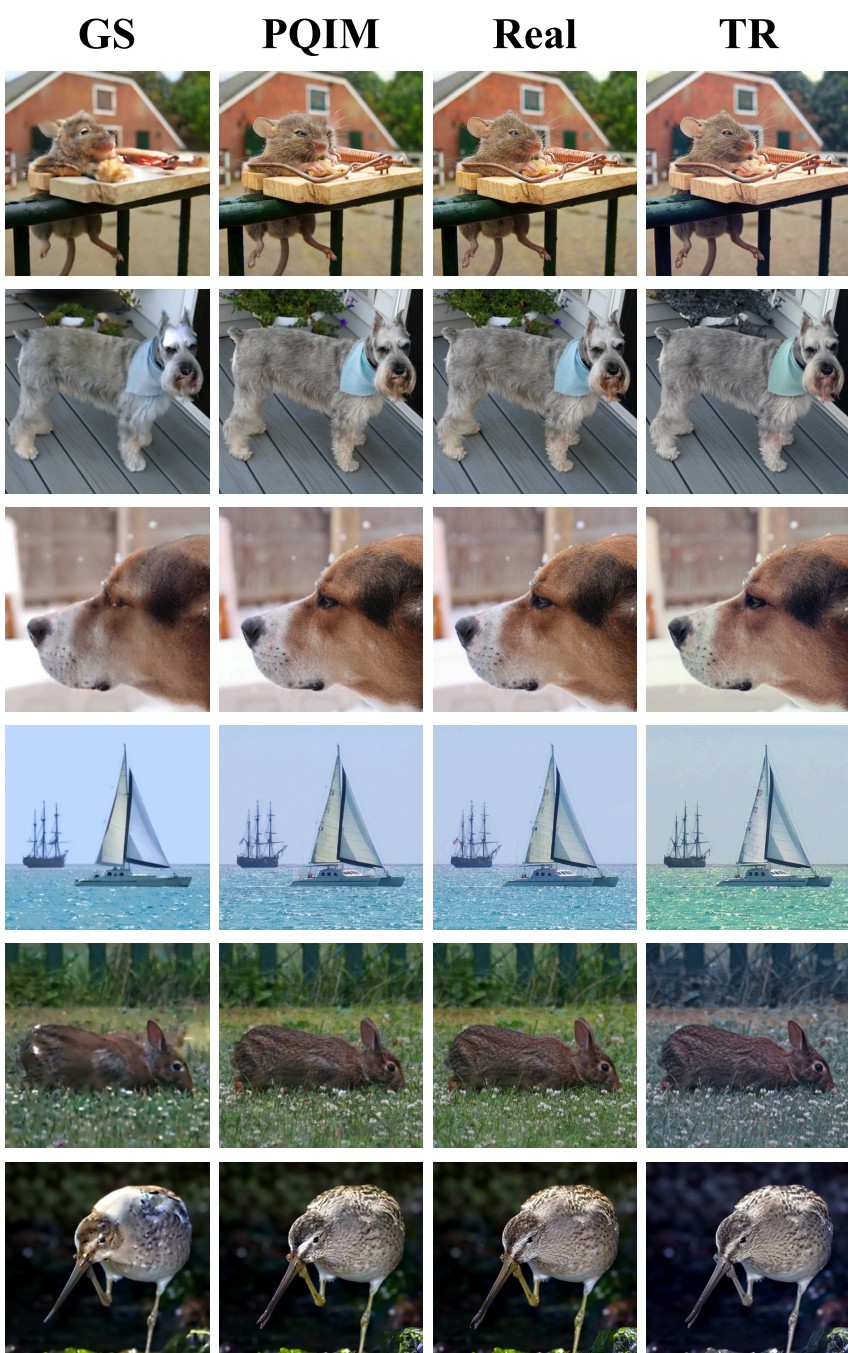

Figure 8: Extended qualitative results for Post-hoc Perceptual Tuning. PQIM successfully embeds the watermark while preserving the original image's color distribution and fine details, contrasting with the artifacts observed in baselines.

**Post-hoc Perceptual Tuning.** Fig. 8 presents qualitative comparisons for the post-hoc perceptual tuning scenario. A critical challenge in post-hoc watermarking is minimizing the perceptual deviation from the source image while embedding a robust signal. As observed, PQIM preserves the intricate texture and global colorimetry of the original images significantly better than baseline methods.

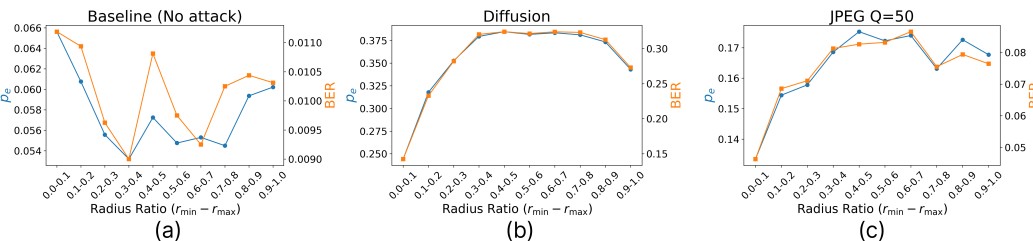

Figure 9: (a) Baseline results of $p_e$, $BER$, (b) $p_e$ and $BER$ under Regeneration Attack, (c) $p_e$ and $BER$ under JPEG @ $Q = 50$

### A.7 FAILURE CASE.

To probe the perceptual limits of phase-only modulation, we conduct a stress test in which PQIM is operated far beyond the standard message regime used in the main experiments in Sec. 4. Concretely, we gradually increase message lengths so that the watermark message occupies most of the latent frequency space. For each setting, we generate images on the same prompt set and visually compare. (1) unwatermarked image, (2) PQIM at the default message length used in the main paper and (3) PQIM under the extreme messages length.

Fig. 10 shows representative examples from this stress test. Under the extreme message lengths, the text prompt is still broadly respected, but we observe that the depicted scene can change, become noticeably simpler and details are gone. In particular, cluttered or texture-heavy backgrounds are often collapsed into smoother regions, secondary objects disappear, and object layouts can be rearranged. Thus, while PQIM remains imperceptible in the conservative operating regime used for all main experiments, these stress-test failures illustrate that sufficiently aggressive phase modulation can alter scene semantics even when the prompt itself appears to be satisfied.

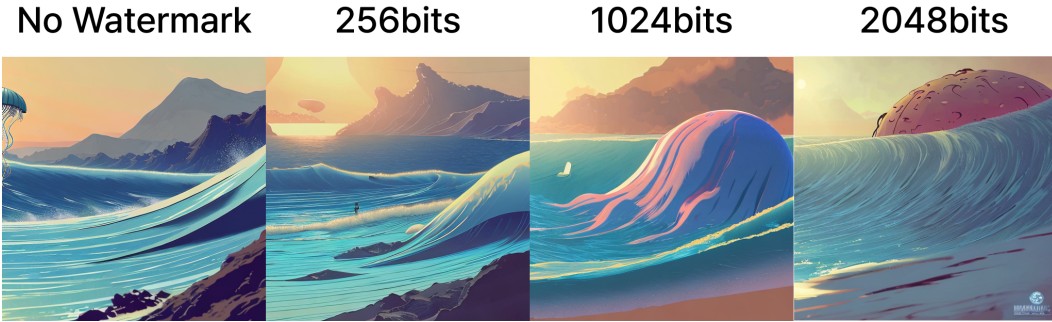

Figure 10: Extended qualitative results for Generative Watermarking. The images demonstrate that embedding the watermark into the initial noise phase preserves the generative quality and semantic coherence without introducing visible artifacts. One representative example uses the complex prompt "A very beautiful serene coastal landscape scene with a GIANT MECHA JELLYFISH looming in the distance, bright sunny waves splashing on the beach, The Great Wave off Kanagawa by Hokusai, translucent, rendered by Simon Stålenhag, Beeple, Makoto Shinkai, Syd Mead, WLOP, 4K UHD, octane render".

### A.8 CLASSICAL DISTORTIONS EXAMPLES.

Fig. 11 shows the examples of the classical distortions we applied. As we can see Gaussian noise and salt and pepper noise disrupts the structure of the image, so that it leads to the degradation of PQIM performance since we utilize the phase which contains the structural information of the image.

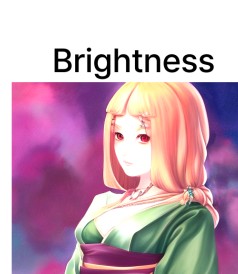
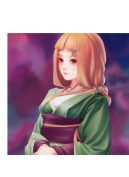
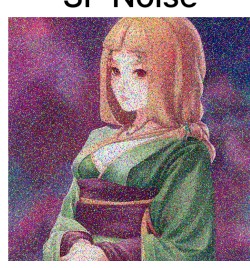

Figure 11: Visual examples of classical distortions that are applied in the main experiments.

## B  SECURITY ANALYSIS & IMPLEMENTATION DETAILS

In this appendix we formalize the cryptographic pipeline underlying PQIM and provide a WM-IND-CPA indistinguishability theorem for our key- and nonce-dependent subspace selection. Our analysis follows the standard pseudorandom generator (PRG) framework used in modern cryptography.

### B.1  CRYPTOGRAPHIC PIPELINE: KDF, CSPRNG, AND NONCE

A user provides a passphrase $\mathcal{P}$. We derive a 256-bit secret key using SHA-256:

$$K = \text{SHA-256}(\mathcal{P}) \in \{0,1\}^{256}. \tag{9}$$

This key $K$ is used only to generate secret embedding locations.

For each image, we sample a unique per-image nonce $N \in \{0,1\}^{\lambda_N}$. By using a unique per-image nonce and assuming that HMAC-DRBG is a secure PRG, the resulting location streams $S = F(K, N)$ are computationally indistinguishable from independent uniform strings. Thus, no statistical correlation, linkability grows across different images generated with the same $K$. We denote its output by

$$S = F(K, N) := \text{HMAC-DRBG}_{\text{SHA256}}(K, N) \in \{0,1\}^{L}, \tag{10}$$

where $L$ is the number of bits required to generate all embedding locations.

The bitstream $S$ is then mapped to mid-frequency indices in the frequency domain. Let $\mathcal{C}$ be the set of candidate mid-frequency coefficients, and let $M = |\mathcal{C}|$. We interpret $S$ as a source of randomness that induces a permutation $\pi$ over $\mathcal{C}$. We take the first $LN_{\text{bins}}$ elements of this permutation and split them into $L$ disjoint subsets

$$\Omega_1, \ldots, \Omega_L \subseteq \mathcal{C}, \quad \Omega_k \cap \Omega_{k'} = \emptyset \text{ for } k \neq k'. \tag{11}$$

Each $\Omega_k$ is the key- and nonce-dependent subset of coefficients used to embed the $k$-th message bit $b_k$ via phase-only QIM, as described in the main text.

### B.2  PRG SECURITY

We first recall the standard PRG security definition.

**Definition 1** (PRG Security). *Let $F : \mathcal{K} \times \mathcal{N} \to \{0,1\}^L$ be a keyed pseudorandom generator. For any probabilistic polynomial-time (PPT) distinguisher $\mathcal{D}$, define*

$$\mathbf{Adv}_{\mathcal{D}}^{\mathrm{PRG}}(\lambda) = \left| \Pr\left[\mathcal{D}(F(K,N)) = 1\right] - \Pr\left[\mathcal{D}(r) = 1\right] \right|, \tag{12}$$

*where $K \leftarrow \{0,1\}^\lambda$ is a uniformly random key, $N$ is a nonce, and $r \leftarrow \{0,1\}^L$ is a uniformly random string. We say that $F$ is a secure PRG if $\mathbf{Adv}_{\mathcal{D}}^{\mathrm{PRG}}(\lambda)$ is negligible in the security parameter $\lambda$ for all PPT distinguishers $\mathcal{D}$.*

We now define a security game specialized to PQIM. Let $\lambda$ be the security parameter and let $G : \mathcal{Z} \to \mathcal{X}$ denote the diffusion sampler mapping an initial latent $z$ to an image $x$. Let $E(z, b, S)$ denote the PQIM embedding function that takes a latent $z$, a message $b \in \{0,1\}^L$, and a location bitstream $S$ and produces a watermarked latent $z'$.

**Definition 2** (Mechanism 1: Cryptographic Indistinguishability of Subspaces). *Fix a diffusion model $G$ and message length $L$. The security game proceeds as follows:*

1. *The challenger samples a secret key $K \leftarrow \{0,1\}^\lambda$ and chooses a random bit $b^\star \leftarrow \{0,1\}$.*

2. *The adversary $\mathcal{A}$ may specify any side information, e.g., prompts or conditions for the diffusion model. For simplicity, we denote the resulting latent distribution by $\mathcal{Z}$.*

3. *The challenger samples a nonce $N$ and a random message $b \leftarrow \{0,1\}^L$, and draws $z \sim \mathcal{Z}$.*

4. *If $b^\star = 0$ (PRG world), the challenger computes*

$$S = F(K, N),$$

   *whereas if $b^\star = 1$ (random world), the challenger samples*

$$S \leftarrow \{0,1\}^L.$$

5. *The challenger computes the watermarked latent*

$$z' = E(z, b, S)$$

   *and generates the image*

$$X = G(z').$$

6. *The adversary $\mathcal{A}$ receives $X$ and outputs a bit $\hat{b}$ as its guess for $b^\star$.*

*We define the advantage of $\mathcal{A}$ as*

$$\mathbf{Adv}_{\mathcal{A}}^{\mathrm{WM}}(\lambda) = \left| \Pr\left[\hat{b} = 1 \mid b^\star = 1\right] - \Pr\left[\hat{b} = 1 \mid b^\star = 0\right] \right|. \tag{13}$$

Intuitively, the only difference between the two worlds is how the embedding locations, which are the frequency subspaces $\Omega_k$, are chosen. Either via the PRG $F(K, N)$ or via a truly uniform random string.

### B.3 SECURITY OF PQIM POSITIONS

We now state and prove our main security theorem for the cryptographic subspace selection in PQIM.

**Theorem 2** (Security of PQIM Positions). *Let $F(K,N) = \mathrm{HMAC\text{-}DRBG}_{\mathrm{SHA256}}(K,N)$ be a secure PRG in the sense of Definition 2. Then for any PPT adversary $\mathcal{A}$ in the above game for PQIM there exists a PPT distinguisher $\mathcal{D}$ for the PRG such that*

$$\mathbf{Adv}_{\mathcal{A}}^{\mathrm{WM}}(\lambda) = \mathbf{Adv}_{\mathcal{D}}^{\mathrm{PRG}}(\lambda). \tag{14}$$

*In particular, if $F$ is a secure PRG, then $\mathbf{Adv}_{\mathcal{A}}^{\mathrm{WM}}(\lambda)$ is negligible for all PPT adversaries $\mathcal{A}$.*

*Proof.* We construct a PPT distinguisher $\mathcal{D}$ for the PRG $F$ using any given PPT adversary $\mathcal{A}$.

**Construction of $\mathcal{D}$.** The distinguisher $\mathcal{D}$ is given a challenge string $S^\star \in \{0,1\}^L$ which is either $F(K, N)$ for a random key $K$ and nonce $N$ or a uniform random string $r$. Its goal is to decide which is the case.

$\mathcal{D}$ simulates the watermarking environment for $\mathcal{A}$ as follows:

1. $\mathcal{D}$ forwards any side information queries of $\mathcal{A}$ to the diffusion model setup, obtaining a latent $z \sim \mathcal{Z}$.

2. $\mathcal{D}$ samples a random message $b \leftarrow \{0,1\}^L$.

3. $\mathcal{D}$ computes a watermarked latent
$$z' = E(z, b, S^\star),$$
using the challenge string $S^\star$ as the location bitstream that defines the subspaces $\Omega_1, \ldots, \Omega_L$.

4. $\mathcal{D}$ generates the image
$$X_{S^\star} = G(z').$$

5. $\mathcal{D}$ feeds $X_{S^\star}$ to $\mathcal{A}$ and receives the output bit $\hat{b}$.

6. $\mathcal{D}$ outputs 1 if $\hat{b} = 1$ and 0 otherwise.

**Analysis.** We consider two cases:

**Case 1:** $S^\star = F(K, N)$ **(PRG world).** In this case, the distribution of $S^\star$ is identical to that used in the game when $b^\star = 0$. Therefore, the joint distribution of $(z, b, S^\star, z', X_{S^\star})$ as seen by $\mathcal{A}$ is exactly the same as in the $b^\star = 0$ branch of the above game. Hence,
$$\Pr\left[\mathcal{D}(F(K, N)) = 1\right] = \Pr\left[\mathcal{A}(X) = 1 \mid b^\star = 0\right]. \tag{15}$$

**Case 2:** $S^\star = r$ **(random world).** Similarly, when $S^\star$ is a uniform random string $r \leftarrow \{0,1\}^L$, the distribution of $(z, b, S^\star, z', X_{S^\star})$ matches exactly the $b^\star = 1$ branch of the above game, where embedding locations are chosen from a truly uniform bitstring. Therefore,
$$\Pr\left[\mathcal{D}(r) = 1\right] = \Pr\left[\mathcal{A}(X) = 1 \mid b^\star = 1\right]. \tag{16}$$

**Equating advantages.** By the definition of $\mathbf{Adv}_{\mathcal{D}}^{\mathrm{PRG}}(\lambda)$ and $\mathbf{Adv}_{\mathcal{A}}^{\mathrm{WM}}(\lambda)$, we have
$$\mathbf{Adv}_{\mathcal{D}}^{\mathrm{PRG}}(\lambda) = \left|\Pr\left[\mathcal{D}(F(K, N)) = 1\right] - \Pr\left[\mathcal{D}(r) = 1\right]\right| \tag{17}$$
$$= \left|\Pr\left[\mathcal{A}(X) = 1 \mid b^\star = 0\right] - \Pr\left[\mathcal{A}(X) = 1 \mid b^\star = 1\right]\right| \tag{18}$$
$$= \mathbf{Adv}_{\mathcal{A}}^{\mathrm{WM}}(\lambda). \tag{19}$$

Thus any non-negligible advantage for $\mathcal{A}$ immediately yields a non-negligible PRG advantage for $\mathcal{D}$, which contradicts the assumption that $F$ is a secure PRG. Therefore, under the PRG assumption, $\mathbf{Adv}_{\mathcal{A}}^{\mathrm{WM}}(\lambda)$ must be negligible for all PPT adversaries $\mathcal{A}$. $\qquad\square$

**Implication on Linkability** The theorem proves that the embedding subspaces behave like truly random subsets for each image. Consequently, an adversary cannot exploit the reuse of key $K$ to link different images, as determining whether two images share the same key is at least as hard as distinguishing the PRG output from random noise.

### B.4 Mechanism 2: Bounded Perturbation & Sampler Stability

While Mechanism 1 focuses on the cryptographic indistinguishability of the key- and nonce-dependent embedding subspaces, we also need to argue that the magnitude of the perturbation introduced by PQIM is small for any efficient detector to exploit. In this section, we derive an upper bound on the latent space distortion and combine it with known smoothness properties of diffusion samplers.

**1. Per-coefficient perturbation bound.** Let $z \in \mathbb{C}^D$ denote the frequency-domain latent vector and let $\Omega \subseteq \{1, \ldots, D\}$ be the set of modified coefficients, with sparsity

$$\rho = \frac{|\Omega|}{D}.$$

For $k \in \Omega$, write $z_k = |z_k|e^{i\phi_k}$ and $z'_k = |z_k|e^{i\phi'_k}$, where $\phi'_k$ is obtained by phase QIM. By construction of the two interleaved codebooks $Q_0$ and $Q_1$ with step size $\Delta$, the quantization error satisfies

$$|\delta\phi_k| = |\phi'_k - \phi_k| \leq \frac{\Delta}{2} \quad \text{for all } k \in \Omega.$$

The complex-plane geometry then yields

$$|z'_k - z_k| = 2|z_k|\,\big|\sin(\delta\phi_k/2)\big| \leq 2|z_k| \cdot \frac{|\delta\phi_k|}{2} \leq |z_k|\frac{\Delta}{2}, \tag{20}$$

where we used $\sin x \leq x$ for $x \geq 0$. Thus

$$\|z' - z\|_2^2 = \sum_{k\in\Omega} |z'_k - z_k|^2 \leq \left(\frac{\Delta}{2}\right)^2 \sum_{k\in\Omega} |z_k|^2. \tag{21}$$

Assuming the latent coefficients are approximately i.i.d. with unit variance, as is standard for Gaussian noise latents, we have $\mathbb{E}|z_k|^2 \approx 1$ and hence

$$\mathbb{E}\|z\|_2^2 \approx D, \qquad \mathbb{E}\|z' - z\|_2^2 \leq \left(\frac{\Delta}{2}\right)^2 |\Omega|.$$

Therefore the *expected relative $L_2$ distortion* admits the bound

$$\frac{\mathbb{E}\|z' - z\|_2^2}{\mathbb{E}\|z\|_2^2} \leq \rho\left(\frac{\Delta}{2}\right)^2, \qquad \mathbb{E}\frac{\|z' - z\|_2}{\|z\|_2} \leq \sqrt{\rho}\,\frac{\Delta}{2}, \tag{22}$$

where the second inequality follows from Jensen's inequality.

**2. Specialization to PQIM.** In PQIM, the embedding subspaces are further structured by the distributed encoding across $L$ message bits. Let $N_{\text{bins}}$ denote the redundancy per bit, so that each bit $b_k$ is encoded over a disjoint subset $\Omega_k \subseteq \Omega$ with $|\Omega_k| = N_{\text{bins}}$, and the sets $\Omega_1, \ldots, \Omega_L$ form a partition of $\Omega$. Hence

$$|\Omega| = \sum_{k=1}^{L} |\Omega_k| = LN_{\text{bins}}, \qquad \rho = \frac{|\Omega|}{D} = \frac{LN_{\text{bins}}}{D}.$$

Substituting this into equation 22 shows that the latent perturbation is controlled jointly by the redundancy $N_{\text{bins}}$, the message length $L$, the latent dimension $D$, and the quantization step $\Delta$. In all our experiments, we choose these hyperparameters such that $LN_{\text{bins}} \ll D$, i.e., only a small fraction of coefficients are modified in a high-dimensional latent, and the bound in equation 22 remains a moderate constant. Empirically, the resulting watermarked images are visually indistinguishable from clean generations and exhibit small or no degradation in PSNR/SSIM/LPIPS (see Tab. 1), indicating that the actual distortion is smaller than this worst case bound.

**3. Smoothness of diffusion samplers.** A large body of work on diffusion models and generative networks shows that the sampling map from the initial noise to the final image behaves smoothly with respect to input perturbations, both theoretically and empirically Song et al. (2025). While we do not explicitly bound a global Lipschitz constant for a specific model, these results support the modeling assumption that small perturbations in the initial noise latent lead to small, smooth changes in the generated image. Combined with the latent distortion bound in equation 22 and our empirical measurements in Tab. 1 and Tab. 3, this suggests that PQIM keeps watermarked images within the natural variability of the model's output distribution, making detection difficult in practice.

