# OpenReview forum: "Hiding in the Phase: A Provably Robust Watermark for Diffusion Models"
_ICLR.cc/2026/Conference — Submitted to ICLR 2026_

### Official Review · Reviewer_gnUq · 2025-10-30

**Soundness:** 3
**Presentation:** 3
**Contribution:** 3
**Rating:** 8
**Confidence:** 3

**Summary:**

This paper proposes Phase-Quantization Invisible Marking (PQIM), a robust and provably secure watermarking method for diffusion models. Addressing the vulnerability of existing techniques that use uniform, predictable embedding patterns (which are prone to targeted removal attacks), PQIM leverages a cryptographic key to pseudo-randomly select regions in the phase spectrum of the noise latent for watermark embedding. This structural heterogeneity makes the watermark difficult to locate or remove without the key. The approach guarantees reliability through information-theoretic analysis and preserves image quality by restricting modifications to mid-frequency phase components. Extensive experiments demonstrate that PQIM outperforms previous state-of-the-art in both robustness to common and advanced attacks and in maintaining high perceptual fidelity.

**Strengths:**

**Originality:**

PQIM presents a novel phase-based watermarking strategy for diffusion models, making use of cryptographic keys to ensure robustness and security against targeted attacks.

**Quality:**

Methodology is thorough, with theoretical guarantees and extensive experiments showing strong robustness and fidelity compared to prior approaches.

**Clarity:**

The paper explains the motivations, methods, and findings with clear logic supported by diagrams and empirical results.

**Significance:**

PQIM advances watermarking in generative AI by offering a training-free, provably secure, and high-fidelity solution, which is timely for increasing demands in content authentication and provenance.

**Weaknesses:**

1. White-box Extraction Requirement. From the Section 3.2 (Watermark Extraction), Conclusion, the extraction protocol currently relies on DDIM inversion using access to the original diffusion model. This white-box assumption may not suit real-world use when the model weights or architecture are inaccessible. Extending to black-box recovery or ensemble approaches would broaden usability.
2. While PQIM is robust to most distortions and attacks, it struggles under strong Gaussian and salt-and-pepper noise, with detection rates and bit accuracy dropping significantly. Improving noise resilience through advanced decoding or training is an unresolved challenge highlighted in quantitative tests.
3. PQIM’s ability to embed long messages is limited by the number of frequency bins available and the redundancy needed for robustness. When using the largest payloads (e.g., 512 bits), the bit accuracy and reliability gradually degrade, which constrains applications needing large metadata or cryptographic tags.

**Questions:**

1. Can the authors elaborate on possible improvements or alternative encoding strategies to increase the maximum recoverable payload size, while maintaining robustness and fidelity? Are there practical ways to enable higher capacity for real-world metadata or secure tags beyond the tested 512 bits?
2. Is it possible to implement watermark extraction in scenarios where the diffusion model is inaccessible (“black-box” settings), or via model-agnostic techniques? If so, what future directions or adaptations could extend PQIM to broader practical contexts, especially for proprietary models?
3. Given the notable reduction in watermark accuracy under strong Gaussian or salt-and-pepper noise (Appendix A.4, Table 3), could the authors propose further improvements in noise-aware decoding or preprocessing? Would retraining or adversarial robustness techniques meaningfully boost reliability for these edge cases?

---

> ### Author Response · Authors · 2025-11-23
> **Response to Reviewer gnUq**
>
> Thank you very much for your careful review of our paper and thoughtful comments. We appreciate with your positive feedback on the **originality of our key-dependent phase watermarking strategy**, the **thoroughness of our methodology with theoretical guarantees** and the **significance of PQIM** in advancing training-free and provably secure solutions.
> We hope the following responses can help clarify the scalability of our method and address your concerns regarding payload capacity.
>
> ---
>
> **Q1: Can the authors elaborate on possible improvements or alternative encoding strategies to increase the maximum recoverable payload size (beyond 512 bits), while maintaining robustness and fidelity?**
>
> **A1:** Thank you for this insightful question. As we discussed with another reviwer, we acknowledge that the performance degradation observed as the payload increases in our experiments is primarily a **spatital bandwidth constraint** stemming from the small latent resolution ($64 \times 64$) of the Stable Diffusion v1.4-2.1 models, rather than a fundamental limitation of the PQIM framework.
> To scale beyond 512 bits for real-world applications, we propose a two-fold strategy: (1) leveraging the **spatial scalability** of modern architectures and (2) adopting **high-efficiency encoding** strategies.
>
> 1. **Spatial Scalability**: Expanding bandwidth via modern architectures. The most direct improvement comes from transitioning to modern models like SDXL or Stable Diffusion 3 (SD3), which utilize a larger latent space.
>     - **Higher Capacity Gain:** This increase in latent resolution provides more frequency bins compared to the models tested. Theoretically, this implies that we can embed more payload bits into an SDXL latent while maintaining the same redundancy ($N_{bins}$) and formal robustness guarantee established in Theorem 1.
>     - Table (TPR@1%FPR/Bit Acc.)
>
>
>         | **Attack Type** | **Parameter** | **SD v2.1 (1024 bits)** | **SDXL (1024 bits)** | **Bit Acc. Gain** |
>         | --- | --- | --- | --- | --- |
>         | **No Distortion** | - | 1.0000 / 0.9397 | **1.0000 / 1.0000** | +6.02% |
>         | **JPEG** | $Q=25$ | 1.0000 / 0.8417 | **1.0000 / 0.9496** | +10.79% |
>         | **Gaussian Blur** | $r=5$ | 1.0000 / 0.7396 | **1.0000 / 0.8485** | +10.89% |
>         | **Gaussian Noise** | $std=0.1$ | 0.9760 / 0.6552 | **0.9980 / 0.7777** | +15.75% |
>         | **Brightness** | $\times 2.0$ | 0.9980 / 0.9065 | **1.0000 / 0.9955** | +8.90% |
>         | **Resize** | $0.5\times$ | 1.0000 / 0.9165 | **1.0000 / 0.9992** | +8.27% |
>         | **SP Noise** | $prob=0.2$ | 1.0000 / 0.7095 | **1.0000 / 0.7310** | +2.94% |
>     - **Universality via Invertibility:** This expansion is applicable regardless of the underlying generative mechanism (e.g., flow matching in SD3). Since PQIM only requires an invertible path to estimate the initial noise, it can universally apply to any deterministic sampling architecture, instantly unlocking this expanded capacity.
> 2. **Alternative Encoding Strategy**: Applying high-efficiency error-correcting codes (ECC) & Soft-Decision. To utilize this expanded capacity for practical, error-free metadata storage, we propose replacing the current  majority-vote repetition code, which has a code rate $R \approx 1/20$ in our experiments, with modern ECC.
>     - **Spectral Efficiency & Coding Gain:** The current majority voting scheme relies on high redundancy (code rate $R \approx 1/20$), which is spectrally inefficient. By transitioning to modern ECCs (such as LDPC or Polar codes) that operate near the Shannon limit, we can achieve comparable robustness with a significantly higher code rate.
>     - **Soft-Decision Decoding:** Furthermore, upgrading to soft-decision decoding which uses log-likelihood ratios instead of hard thresholding provides an additional SNR gain. This ensures that even if the payload is increased, the system remains resilient to strong distortions like Gaussian noise where hard-decision decoders might fail.

---

> ### Author Response · Authors · 2025-11-23
>
> **Q2: Is it possible to implement watermark extraction in scenarios where the diffusion model is inaccessible (“black-box” settings)? If so, what could be the future directions or adaptations?**
>
> **A2:** Thank you for this forward-looking question. To address this, we first clarify the primary deployment scenario of PQIM and then discuss how verification can still be performed when direct access to the original diffusion model is restricted.
>
> **1. Primary Scenario: Provenance Tracking by Model Owners (White-box)**
>
> Standard generative watermarking schemes are typically deployed by model owners (e.g., API service providers) to protect intellectual property and track misuse.
>
> - **Operational Context:** In this setting, the same entity generates images and later verifies them using a secret watermark key. The extraction procedure is explicitly keyed: given the secret key and nonce, the verifier reconstructs the frequency-bin selection and decodes the embedded payload from the recovered latent.
> - **Practicality:** Since the model resides on the provider's server, maintaining white-box access for the verification module is seamless and does not hinder scalability.
>
> **2. Extended Scenario: Black-box Extraction via Proxy Model**
>
> We also consider scenarios where third-party verifiers need to check watermark presence but do not have access to the proprietary generation model. In such cases, the model owner can share the secret watermark key and the per-image nonce with the verifier, while keeping the original model weights private. The verifier then uses a compatible **proxy model** to approximate the inversion path.
>
> - **Feasibility Test:** We evaluated this scenario by generating watermarked images with Stable Diffusion v2.1 and performing extraction using Stable Diffusion v2.0 as a publicly available proxy model on the verifier side. The verifier knows the watermark key and the nonce for each image, and therefore can reconstruct the same frequency-bin pattern used at embedding time, but only has access to the proxy model, not the original one.
> - **Results:** Despite the parameter mismatch between SD v2.1 (generation) and SD v2.0 (proxy), the shared VAE and latent semantics allow PQIM to retain strong detection performance
>
>
>     | **Attack Type** | **Parameter** | **TPR@1%FPR** | **Average Bit Accuracy** |
>     | --- | --- | --- | --- |
>     | **No-Distortion** | - | 1.0000 | 1.0000 |
>     | **JPEG** | $Q=25$| 1.0000 | 0.9572 |
>     | **Gaussian Blur** | $r=5$ | 1.0000 | 0.8559 |
>     | **Gaussian Noise** | $std=0.1$ | 0.9920 | 0.7869 |
>     | **Brightness** | $\times 2.0$ | 0.9980 | 0.9946 |
>     | **Resize** | $0.5\times$ | 1.0000 | 0.9997 |
>     | **SP Noise** | $prob=0.2$ | 1.0000 | 0.7387 |
> - **Implication:** This experiment shows that PQIM remains practically verifiable even when the exact generation model is not available: an authorized verifier with the secret key/nonce pair can still rely on a closely related, parameter-mismatched model from the same family to recover the watermark with high accuracy. We see this as one of the proxy scenarios (e.g., a proprietary fine-tuned model vs. a publicly released base model).

---

> ### Author Response · Authors · 2025-11-23
>
> **Q3: Could the authors propose further improvements in noise-aware decoding or preprocessing? Would retraining or adversarial robustness techniques meaningfully boost reliability?**
>
> **A3:** Thank you for this suggestion. We agree that retraining or adversarial fine-tuning could theoretically boost reliability by forcing the model to learn robust watermark representations. However, such approaches sacrifice the primary advantage of PQIM, plug-and-play and zero-trianing cost.
>
> Also, we carefully considered preprocessing as a potential solution. However, strictly applying spatial filtering in a blind setting risks altering the high-frequency phase information essential for PQIM, potentially degrading detection in clean images.
>
> Instead, we argue that the most effective solution lies in the noise-robust decoding strategies we proposed in **A1** (High-Efficiency ECC & Soft-Decision).
>
> **1. Soft-Decision Decoding as intrinsically Noise-Robust Approach**
> The performance drop under Gaussian or S&P noise occurs because hard decision decoding forces a binary choice on corrupted, unreliable frequency bins.
>
> - **Mechanism:** By adopting soft-decision decoding (as detailed in A1), the decoder calculates Log-Likelihood Ratios (LLR).
> - **Noise Adaptation:** In the presence of strong noise, the LLR magnitudes for corrupted bins naturally shrink towards zero, indicating uncertainty. The decoder automatically down-weights these noisy bins and relies on the preserved, high-confidence bins. This is, by definition, a noise-aware mechanism that adapts per image without manual intervention.
>
> **2. Channel Coding (ECC) to Bridge the SNR Gap**
>
> - **Coding Gain:** The integration of ECC provides a significant coding gain. This gain allows the system to tolerate a much lower Signal-to-Noise Ratio (SNR).
> - **Result:** Even if Gaussian or S&P noise corrupts a significant portion of the spectrum, the ECC can correct these errors using the redundancy distributed across the phase spectrum.

---

### Official Review · Reviewer_p1ZU · 2025-11-01

**Soundness:** 2
**Presentation:** 1
**Contribution:** 2
**Rating:** 2
**Confidence:** 3

**Summary:**

This paper showed an important weakness in current semantic image watermarking schemes and proposed a new watermarking scheme for Stable Diffusion models as an alternative to the current PRC, Gaussian Shading, etc. These current schemes in semantic watermarking all share one common vulnerability: dependence on systematic and uniform structural rules. Each semantic scheme, depending on its specific structure, impose different types of susceptibilities, leaving possibility for a developed attack surface.

The authors provide a novel alternative, PQIM, which writes bits into the phase of the initial noise while keeping the amplitude statistics the same, ensuring the latent and image quality are preserved. They use a secret key to pick a sparse set of frequency locations, where each bit is spread across many of them. These bits are then read back by DDIM inversion with a quantize-and-vote decoder.

Key results: PQIM beats Tree-Ring and Gaussian Shading on most attacks (JPEG, blur, resize, regeneration, learned codecs) while keeping image quality, and the watermark survives under a targeted latent-perturbation test; the few cases where others do better are heavy Gaussian or salt-and-pepper noise, where GS/Tree-Ring keep higher TPR, but PQIM still recovers bits. Their claim is that overall, this scheme outperforms most existing schemes across a broad range of advanced attacks, although some specific schemes may outperform in some specific attacks.

Main contributions: a keyed secret subspace that avoids uniform structure, a phase-only QIM encoder with distributed redundancy and quantize-and-vote decoding, plus an analysis showing the PQIM bit error rate drops exponentially with more redundancy. They also provided a perceptual watermark version (a watermarker that hides a whole bit message rather than 1/0 bit, i.e. a stenography scheme).

The authors conclude that global, uniform structured schemes expose a predictable attack, whereas keyed phase-only subspace shrinks the predictable attack surface.

**Strengths:**

There are some strengths of this paper. They do provide a functional, mostly secure, watermarking scheme, with satisfactory quality results. Their claim that current watermarkers have a uniform, predictable attack surface appears solid and aligns with their reasoning.

In terms of quality, PQIM appears to perform quite well, clearly outputting the perceptually closest result to the original. The same is apparent with the perceptual tuning scheme.
(However, I would like to specify the following: PQIM having a closer prompt to original is due to how it perturbs the generation pipeline; other schemes perturb the generation more, such as PRC where the watermark is embedded into the generation token. Contrary to what the paper implies, resembling the original generated image is not the goal of AI image generation. As long as it matches the prompt with quality, it succeeds.)

Main contributions: a keyed, phase-only QIM watermark for diffusion latents, a simple quantize-and-vote decoder, a constraint on bit error, and a proof that more redundancy lowers BER. They also provided insight to a fundamental weakness regarding current watermarking schemes. The authors devised an acceptable scheme, but QIM-style embedding and majority-vote guarantees are already recognized. The same goes for invert-and-detect pipelines, which are already used in Tree and GS.

**Weaknesses:**

I did not find their claims about its “superior consistency” and “consistently outperforms existing methods” to be conclusive enough. Firstly, for the basic attacks, the results shown in the comparative does not show the scheme to conclusively or continually outperform the best of the other schemes. It seems they show two contradicting tables. According to the table 1 in the main paper, PQIM doesn’t have a TPR @ 1%FPR rate less than 90%. However, we can see that it performs inadequately across many a few different attacks when compared to GS (Gaussian Shading) or Tree (Tree-Ring), especially Gaussian and SP Noise, in table 3 in the appendix. The authors also state that PQIM lacks in these forms of attacks, showing that it does not have superior consistency nor consistently outperforming other methods. Although this isn’t as significant of a margin, in section A.5 of the appendix, they show it is outperformed by Gaussian Shading and Tree-Ring in two out of three stable diffusion attacks.
Thus, paper’s claim of “consistently outperforms” does not hold across all attacks (beaten by GS/Tree-Ring on heavy Gaussian, salt-and-pepper noise, and on some SD-style transforms), so superiority depends on the setting. The scheme’s evaluation breadth could have been wider and should have included benchmarks (e.g., W-Bench, WAVES), to the other watermarkers. There is no undetectability or watermarking CPA guarantee (WM-IND$-CPA), which is essential and provided in major schemes (PRC, GS, Tree-Ring, etc). The claim “no specific weak point” isn’t shown cryptographically and formally.

There are many potential ways to improve the presentation of their claims, reasoning, and results. I noticed there were some clear mistakes, including a sentence copy pasted an extra time, table 5 and 6 being identical copies, and many more clear writing errors, and a random figure displayed at the end without explanation.
In terms of a visual comprehension, the figures are difficult to understand and could have more explanation behind them. I also thought another figure demonstrating the PQIM scheme in a different manner would be really helpful for readers to gain a better grasp.
The core ideas are there but the write-up is a little difficult to follow. I felt key topics, including phase-only embedding, DDIM inversion, and the decoder, could have been further elaborated with more explanation, better introduction, and possibly a theoretical example. The threat model and decode rule, including ties, are not clearly stated, and notation seems inconsistent in some parts. Ideally, a short overview, a theorical example, and another figure would be great for presentation.

**Questions:**

1) How does PQIM perform in benchmarks, possibly W-bench and WAVES, against Tree-Ring, Gaussian Shading, PRC, etc.?

2) Is the watermark fully undetectable (at least WM-IND$-CPA secure)?

3) Does linkability grow if the same key is used across many different prompts or users? Is there a unlinkability guarantee for key and nonce resuse, if it doesn’t provide a per-image nonce?

---

> ### Author Response · Authors · 2025-11-28
> **Response to Reviewer p1ZU**
>
> We would like to express our sincere gratitude for your thorough and constructive feedback. Your detailed review has been instrumental in identifying critical areas for improvement, particularly regarding the clarity of our claims, the presentation quality, and the formal security guarantees.
>
> We deeply appreciate the time and effort you dedicated to reviewing our paper. Your sharp observations regarding the **security proofs** and **presentation issues** were especially valuable.
>
> Based on your insightful suggestions, we have made the following major revisions to our paper:
>
> 1. **Clarification of Claims:** We have refined our claims regardingovov "superior consistency" to clearly distinguish between generative watermarking (robustness focus) and post-hoc perceptual tuning (fidelity focus) tasks, explicitly acknowledging trade-offs in specific noise attacks.
> 2. **Enhanced Presentation:** We have completely overhauled the presentation. This includes removing duplicate tables, redrawing the main pipeline figure for clarity and standardizing notation throughout the paper.
> 3. **Formal Security Analysis:** We have added a formal security analysis in the Appendix, explicitly defining our cryptographic pipeline (KDF + CSPRNG + Nonce) and providing mathematical proofs for WM-IND$-CPA and unlinkability.
> 4. **Expanded Evaluation:** We have incorporated additional benchmarks as suggested.
>
> We hope the following detailed responses and the revised manuscript fully address your concerns.
>
> ---
>
> **Q1: How does PQIM perform in benchmarks, possibly W-bench and WAVES, against Tree-Ring, Gaussian Shading, PRC, etc.?**
>
> **A1** Thank you for pointing out that the evaluation breadth could be wider and for explicitly asking how PQIM behaves under standardized benchmarks compared to Tree-Ring, Gaussian Shading and PRC, etc.
>
> Regarding your suggestion to also include W-Bench, we fully agree that this would be a valuable complement to our evaluation. Unfortunately, a full W-Bench run was not feasible within the rebuttal time and compute budget, especially given the set of new WAVES experiments we have conducted. We therefore prioritized WAVES, which already covers a wide spectrum of classical and learned attacks on a standardized DiffusionDB subset. We view integrating W-Bench as a worthwhile extension and we appreciate your suggestion as a clear direction for future benchmarking work.
>
> - **Experimental Setup** During the rebuttal period, we ran an additional experiment closely following the WAVES protocol. We use 1,000 randomly sampled prompt-image pairs from the DiffusionDB subset (poloclub/diffusiondb) which is used for WAVES benchmark, as a held-out test set and evaluate the generative watermarkers PQIM(Ours), Tree-Ring (TR), Gaussian Shading (GS) and PRC under the standard WAVES attacks (blur, brightness/contrast changes, compression, noise, erasing, diffusion regeneration, ResNet-based adversarial editing, resized crop, rotation and VAE-based compression). Due to the limited time and compute budget, we restricted this benchmark to 1,000 pairs and to generative/semantic watermarking schemes, omitting classical DWT-DCT and DWT-DCT-SVD in this specific experiment. We also followed the WAVES attack configurations and, for each attack, used the mid-strength setting within the parameter range specified by the WAVE benchmark.

---

> ### Author Response · Authors · 2025-11-28
>
> - **Fidelity / Perceptual Quality** Across all methods, visual metrics lie in a very narrow range on the WAVES evaluation set. PQIM, TR and GS are almost indistinguishable in terms of average PSNR/SSIM/LPIPS and aesthetics. In particular, the Watson perceptual error of PQIM is comparable to Tree-Ring and substantially lower than Gaussian Shading, indicating that PQIM does not incur noticeably higher visual distortion than existing semantic watermarkers when evaluated on an external benchmark.
>
>
>     | **Method** | **FID $\downarrow$** | **CLIP $\uparrow$** | **LPIPS  $\downarrow$** | **Watson $\downarrow$** | **PSNR $\uparrow$** | **SSIM $\uparrow$** | **NMI $\uparrow$** | **Aesthetics $\uparrow$** | **Artifcats $\downarrow$** |
>     | --- | --- | --- | --- | --- | --- | --- | --- | --- | --- |
>     | **NO Watermark** | 58.5068 | 33.1252 | 0.768531 | 1.501E+07 | 8.670638 | 0.225787 | 1.017485 | 5.122270 | 2.608365 |
>     | **PQIM** | 58.8116 | 35.0036 | 0.826394 | 1.01E+08 | 8.561328 | 0.192215 | 1.012976 | 5.40887 | 2.434866 |
>     | **GS** | 58.6874 | 34.8539 | 0.824317 | 5.81E+08 | 8.600696 | 0.194198 | 1.013215 | 5.475829 | 2.425683 |
>     | **TR** | 58.7203 | 35.1883 | 0.825493 | 1.05E+08 | 8.695419 | 0.184837 | 1.012505 | 5.494112 | 2.404948 |
>     | **PRC** | 62.6134 | 33.4505 | 0.766707 | 7.02E+06 | 9.348133 | 0.216658 | 1.013832 | 5.157853 | 2.687666 |
>
> - **Robustness under classical distortions and advanced attacks (TPR@1%FPR / Bit Accuracy)** PQIM remains competitive with state-of-the-art semantic watermarkers on the majority of WAVES attacks, despite being training-free with a provable security pipeline.
>
>
>     | **Attack Type** | **Parameter** | **PQIM** | **TR** | **GS** | **PRC** |
>     | --- | --- | --- | --- | --- | --- |
>     | **Blurring** | $\text{kernel size} = 11$ | 0.2590 / 0.5511 | 0.0000 / - | 0.9791 / 0.7033 | 0.0040 / 0.0040 |
>     | **Brightness** | $1.5$ | 1.0000 / 0.9996 | 0.9990 / - | 1.0000 / 0.9946 | 1.0000 / 1.0000 |
>     | **JPEG** | $50$ | 1.0000 / 0.9922 | 0.9540 / - | 1.0000/ 0.9911 | 0.9580 / 0.9585 |
>     | **Contrast** | $1.5$ | 1.0000 / 0.9996 | 0.9970 / - | 1.0000 / 0.9948 | 1.0000 / 1.0000 |
>     | **Regeneration** | $\text{step}=30$ | 1.0000 / 0.9958 | 0.9890 / - | 1.0000 / 0.9936 | 0.9960 / 0.9960 |
>     | **Regeneration** | $\text{step}=60$ | 1.0000 / 0.9915 | 0.9740 / - | 1.0000 / 0.9920 | 0.9740 / 0.9740 |
>     | **Erasing** | $0.125$ | 1.0000 / 0.9998 | 0.9990 / - | 1.0000 / 0.9950 | 1.0000 / 1.0000 |
>     | **Gaussian Noise** | $\sigma=0.05$ | 1.0000 / 0.9869 | 0.9790 / - | 1.0000 / 0.9887 | 0.8760 / 0.8760 |
>     | **Resized & Crop** | $0.75\times$ | 0.0110 / 0.5113 | 0.0060 / - | 0.0600 / 0.5242 | 0.0050 / 0.0045 |
>     | **ResNet18** | $\epsilon = \frac{6}{255}$ | 1.0000 / 0.9870 | 0.9880 / - | 1.0000 / 0.9889 | 0.9210 / 0.9205 |
>     | **Rotation** | $22.5^\circ$ | 0.0070 / 0.5043 | 0.0000 / - | 0.0110 / 0.5033 | 0.0070 / 0.0045 |
>     | **VAE Encoder** | $\text{quality level}=3$ | 1.0000 / 0.9737 | 0.8030 / - | 1.0000 / 0.9782 | 0.7495 / 0.7486 |
>     - Rotation: global rotation of the entire image.
>     - Resized & Crop: random resized cropping of the image and resizing back.
>     - Random erasing (occlusion): masking the image area with a solid patch.
>     - Brightness adjustment: global scaling of intensity.
>     - Contrast adjustment: global contrast scaling.
>     - Gaussian blur: low-pass filtering with a Gaussian kernel.
>     - Additive Gaussian noise: pixel-wise Gaussian noise with standard deviation.
>     - JPEG compression: lossy compression at quality factor.
>     - ResNet-18 feature attack: a PGD adversarial attack in ResNet-18 embedding space, modeling feature-targeted adversarial modifications that preserve human perception but fool downstream models.
>     - VAE-based learned compression: compression via encoder model at specific quality.
>     - Diffusion regeneration: regeneration via a stable diffusion pipeline with noise steps.

---

> ### Author Response · Authors · 2025-11-28
>
> On WAVES, PQIM remains competitive with state-of-the-art semantic watermarkers on the majority of attacks, while exhibiting a similar robustness profile to our main-paper results.
>
> For content-preserving photometric and generative distortions such as brightness/contrast adjustments, JPEG compression, diffusion regeneration, ResNet18-based adversarial edits, and VAE-based learned compression, PQIM achieves TPR@1%FPR = 1.0 with bit accuracy above 0.99, on par with or slightly better than Tree-Ring and GS. Under Gaussian noise (σ = 0.05), PQIM and GS both maintain TPR = 1.0 with very similar bit accuracy (0.9869 vs. 0.9887) and both outperform PRC.
>
> As expected, the most challenging cases are strong low-pass filtering and geometric transforms. Under heavy Gaussian blur (kernel size 11), GS attains higher robustness (0.9791 / 0.7033) than PQIM (0.2590 / 0.5511) and for resized crop (0.75) and global rotation (22.5°) all schemes suffer a dramatic drop in TPR.
>
> The most challenging cases  are the strong geometric transformations, namely rotation and resized crop. Under these attacks, all four generative/semantic watermarking schemes (PQIM, Tree-Ring, GS, and PRC) exhibit very low TPR@1%FPR (close to zero) and bit accuracy around 0.5, indicating essentially random decisions. In other words, PQIM is not uniquely weak here. Rather, these transformations break the spatial/frequency alignment that current semantic watermark designs implicitly rely on.
>
> We therefore interpret rotation and aggressive resized crop as a structural limitation shared by existing semantic watermarkers that embed information at specific coordinates in the generative process, rather than as a PQIM-specific failure mode. Addressing robustness to such strong geometric misalignment likely requires a different class of watermarking mechanisms which we see as an important but orthogonal direction to the present work.

---

> ### Author Response · Authors · 2025-11-28
>
> **Q2 & Q3 Is the watermark fully undetectable (at least WM-IND$-CPA secure)? Does linkability grow if the same key is used across many different prompts or users? Is there a unlinkability guarantee for key and nonce resuse, if it doesn’t provide a per-image nonce?**
>
> A2 & A3 Thank you for these questions regarding the cryptographic integrity of our method. We group them together as they all concern the underlying key/nonce architecture and the detectability and linkability properties of PQIM.
>
> We agree that our original submission did not explain this aspect clearly enough. We will revise both the main paper and the appendix as follows.
>
> 1. **Revision of Main Paper**: In the main text, we will make the description of our cryptographic pipeline more concrete and clear.
>
>     > “frequency bins as secret subspaces using CSPRNG (Cryptographically Secure Pseudorandom Number Generator) initialized with a secret key $K$ and a unique per-image nonce $N$.”
>     >
>
>     The generator outputs pseudorandom bits given a secret key and a public nonce. The actual generator, HMAC-DRBG, will be detailed in the appendix.
>
> 2. **New Appendix Section: Security Analysis & Implementation Details**: We add a new appendix section, “Security Analysis & Implementation Details”, which makes our cryptographic assumptions and guarantees precise and connects them to the implementation (HMAC-DRBG with SHA-256).
>
> Concretely, in this appendix we formalize a WM-IND-CPA-style game for the **key- and nonce-dependent subspace selection**, rather than for the full image distribution. In the game, the adversary must distinguish whether the embedding locations which the frequency subspaces $\Omega_k$) are derived from the PRG $F(K,N)$ or sampled from a truly random source or decide whether two watermarked images share the same key based on their embedding positions. We show that any adversary that can win this game with non-negligible advantage can be turned into an adversary that breaks the pseudorandomness of HMAC-DRBG. Under the standard assumption that HMAC-DRBG with SHA-256 is a secure PRG, this implies that (1) our key/nonce-based embedding subspaces expose no predictable structure beyond what a secure PRG would reveal and (2) reusing the same key across many prompts does not create exploitable linkability because each image uses a fresh per-image nonce. We will clarify in the revised paper that our formal guarantee is at this subspace level. Proving a full image-level WM-IND-CPA guarantee is orthogonal to our focus here and remains an interesting direction for future work.

---

> ### Author Response · Authors · 2025-11-28
>
> We appreciate your detailed critique regarding the presentation and structure of our paper. We have made extensive revisions to improve clarity and added a formal security analysis.
>
> 1. **Weakness: Presentation and Clarity**
>     - **Response:** We have significantly improved the manuscript's presentation based on your suggestions:
>         - **Method Overview:** We added an introductory overview at the **beginning of the Method section** to better guide the reader.
>         - **Figure 1:** We have completely **revised the pipeline in Figure 1** to improve visual clarity and understanding of the proposed framework.
>         - **Experiments Structure:** We **reorganized Section 4** and added explanatory text at the beginning of the section to help readers navigate the experimental setup and results more easily.
>         - **Objective Language:** We have carefully reviewed the text and replaced strong terms like "outperform" with more objective and precise descriptions throughout the paper.
> 2. **Weakness: Threat Model and Security Proof**
>     - **Response:**
>         - **Threat Model:** We have explicitly added a formal **Threat Model in Section 3.2**, defining the adversary's capabilities and knowledge.
>         - **Security Analysis:** To address the lack of formal cryptographic guarantees, we have added a comprehensive **Security Analysis in Appendix B**, including a formal discussion on the security of our key- and nonce-dependent subspace.

---

### Official Review · Reviewer_UeoB · 2025-11-02

**Soundness:** 3
**Presentation:** 2
**Contribution:** 3
**Rating:** 4
**Confidence:** 4

**Summary:**

This paper proposes Phase-Quantization Invisible Marking (PQIM), a novel, training-free semantic watermarking framework for diffusion models that embeds information into the phase spectrum of the initial noise latent. By using a cryptographic key to define a sparse, pseudo-random embedding subspace in the mid-frequency band and applying Quantization Index Modulation (QIM), PQIM ensures imperceptibility while providing provable robustness against attacks.

**Strengths:**

1.	The paper introduces a creative shift from amplitude or spatial manipulations in existing semantic watermarks to exclusive phase-spectrum modulation, achieving a better fidelity-robustness trade-off.
2.	The technical depth is good, with a rigorous theorem grounded in Hoeffding’s inequality proving exponential BER reduction, also empirically validated in Fig.3.
3.	In the context of rising AI-generated misinformation, PQIM advances content provenance tools, with broad applications in authentication and tracking.

**Weaknesses:**

1.	The claimed superior fidelity-robustness trade-off is under-validated for fidelity. Visual results (e.g., Figure 4) are provided, but quantitative metrics are limited to SSIM in Figure 2(d). More comprehensive evaluations, such as CLIP scores for text-image consistency and FID for distributional fidelity, are needed to quantify watermark impact on generated quality. This would strengthen the trade-off claims.
2.	Robustness testing focuses on common distortions (JPEG compression, noise addition, regeneration attacks), but omits geometric operations like rotation, cropping, or scaling，which are common in image editing. It is unclear if PQIM's phase-based design maintains superiority over baselines under these. Including such tests with BER comparisons would better demonstrate practical resilience.
3.	Section 4.2, figure references are sometimes confusing. The citation on line 361 to Figure 3 should specify the subfigure for precision. The reference on line 377 appears to intend Figure 3(c) rather than Figure 2, based on context. Additionally, axis labels (e.g., "radius range" in Figure 2 and "Q step" in Figure 3) do not consistently match descriptions in the main text, potentially confusing readers. Clarifying these would improve clarity.

**Questions:**

1.	The theoretical robustness (Theorem 1) assumes i.i.d Bernoulli errors in phase coefficients, but frequency dependencies in real diffusion process might violate this, potentially overestimating the exponential BER decay. Is it possible to relax this assumption with a more general bound or provide sensitivity analysis on correlation effects in experiments.
2.	The paper's empirical scope is limited to UNet-based Stable Diffusion models, with limited results on newer architectures like DiT-based models. The mid-band embedding is empirically justified, but its effectiveness may not generalize to different noise schedules or latent dimensions. To address this, add cross-model experiments reporting BER and fidelity metrics, enhancing significance.
3.	Given the reliance on phase-only changes, are there scenarios where phase modulation noticeably alters semantics (e.g. in texture-heavy images)? Example of failure case or user studies on perceptibility could help assess real-world imperceptibility.

---

> ### Author Response · Authors · 2025-11-22
> **Response to Reviewer UeoB**
>
> Thank you for your constructive feedback and for acknowledging the soundness and technical depth of our work. We are encouraged by your positive comments on the creative shift to phase-spectrum modulation and the rigorous theoretical grounding of PQIM.
>
> We have carefully considered your suggestions and conducted additional experiments to address your concerns. We hope the following responses clarify the potential misunderstandings and demonstrate the improved quality of our work.
>
> ---
>
> **W1: The claimed superior fidelity-robustness trade-off is under-validated for fidelity. Visual results are provided, but quantitative metrics (CLIP, FID) are limited.**
>
> A1: Thank you for this valuable suggestion. We understand your concern regarding the need for comprehensive quantitative fidelity metrics to substantiate our visual claims.
>
> 1. **Clarification on Figure 2(d):** First, we would like to clarify that Figure 2(d) in the original submission was specifically intended to justify the selection of the starting frequency radius ($r=0.1$) rather than serving as the main fidelity evaluation.
> 2. **Quantitative Evaluation (Post-hoc Watermark):** For the post-hoc watermark which is perceptual tuning task, we have measured perceptual quality, structural similarity and color distribution fidelity against baselines.
>     - Experimental Setup:  We conducted evaluations on 1,000 images generated using the pre-trained Stable Diffusion v2.1. The generation employed DDIM sampling with 50 steps and a classifier-free guidance scale of 7.5, utilizing prompts from the "Gustavosta/Stable-Diffusion-Prompts" dataset.
>
> |  | psnr $\uparrow$ | lpips $\downarrow$ | ssim $\uparrow$ | mssim $\uparrow$ | clip $\uparrow$ | Hist Correl $\uparrow$ | Chi-Square $\downarrow$ | Bhattacharyya $\downarrow$ | Delta E $\downarrow$ | EMD $\downarrow$ |
> | --- | --- | --- | --- | --- | --- | --- | --- | --- | --- | --- |
> | pqim | **28.759423** | 0.1246795 | 0.8775567 | 0.9518686 | **0.9771412** | **0.8911** | **180.6841** | **0.231** | **7.346** | **0.0016** |
> | tree | 25.530289 | **0.0657421** | **0.927096** | **0.9662672** | 0.9752343 | 0.3293 | 7288.426 | 0.6276 | 11.1668 | 0.0028 |
> | gs | 21.1166 | 0.3690025 | 0.7115226 | 0.8281116 | 0.8351717 | 0.4967 | 706.0935 | 0.4848 | 16.6768 | 0.0049 |
>
> Metric Descriptions & Analysis
>
> - **Pixel, Structural and Perceptual Metrics**: PSNR measures the peak signal-to-noise ratio, quantifying absolute pixel-level fidelity. SSIM [1] measures structural degradation, and LPIPS [2] measures perceptual similarity in deep features.
> - **Semantic Fidelity**: CLIP Score [3] measures the semantic alignment between the watermarked image and the prompt.
> - **Color Fidelity Metrics**: To strictly evaluate color preservation, we employed Delta E (CIE76) [4] for perceptual color difference, and histogram-based metrics  including Correlation, Chi-Square, Bhattacharyya, and EMD (Earth Mover's Distance [5]) to measure global distribution shifts.
>
> PQIM achieves the best performance in pixel accuracy, semantic preservation and color fidelity while shows comparable performance in structural scores with baselines. Specifically, PQIM preserves the original color tone and distribution much more effectively than the baselines, validating that our phase modulaton strategy minimizes visual artifacts in the color domain.
>
> 1. **Ongoing Experiments (Generative Watermark)**: Regarding the generative watermark pipeline, we are currently finalizing comprehensive experiments to measure FID and CLIP scores. We are prioritizing this to ensure the reported metrics are statistically rigorous. We commit to providing these quantitative results in the discussion thread shortly within the rebuttal period.
>
> ---
>
> [1] Zhou Wang, Eero P. Simoncelli, and Alan C. Bovik, *“Multiscale Structural Similarity for Image Quality Assessment,”* Proc. Asilomar Conf. on Signals, Systems and Computers, 2003.
>
> [2] Richard Zhang, Phillip Isola, Alexei A. Efros, Eli Shechtman, and Oliver Wang, *“The Unreasonable Effectiveness of Deep Features as a Perceptual Metric,”* CVPR, 2018.
>
> [3] Alec Radford et al., *“Learning Transferable Visual Models From Natural Language Supervision,”* ICML, 2021
>
> [4] Luo, M.R., Cui, G. and Rigg, B. (2001), The development of the CIE 2000 colour-difference formula: CIEDE2000. Color Res. Appl., 26: 340-350.
>
> [5] Rubner, Y., Tomasi, C. & Guibas, L.J. The Earth Mover's Distance as a Metric for Image Retrieval. *International Journal of Computer Vision* 40, 99–121 (2000)

---

> ### Author Response · Authors · 2025-11-22
>
> **Q1: The theoretical robustness (Theorem 1) assumes i.i.d. Bernoulli errors, but real diffusion might violate this. Is it possible to relax this or provide sensitivity analysis?**
>
> A1: We appreciate this insightful comment regarding the theoretical assumptions. We have conducted both a quantitative analysis and a correlation sensitivity analysis to address your concern.
>
> 1. **Empirical Validation of the Generalized Bound:** While real-world diffusion processes may introduce dependencies, our experiments show that the Bit Error Rate (BER) decay follows the theoretical prediction with high precision.
> We relaxed the strict i.i.d. assumption to a generalized bound: $P_B \approx C \cdot \exp(-2 N_{\text{bins}} (0.5 - p_e)^2)$. As illustrated in Fig. 3 (a), the experimental data fits this theoretical model well.
>     - **Regression Analysis:** The fit yields a high **Coefficient of Determination ($R^2$) of 0.95** and a Log-scale RMSE of **0.59**.
>     - **Implication:** The slope which closely matches the theoretical value ($-1$ in the log-log plot of BER vs. $e^{-t}$) confirms that the "correlation effects" do not dampen the exponential convergence rate provided by the redundancy ($N_{bins}$). The correlations merely introduce a constant scaling factor $C$ ($C \approx 0.027$), which does not compromise the asymptotic robustness.
> 2. **Sensitivity Analysis on Correlation Effects:** To directly verify why the theory holds so well despite potential dependencies, we conducted a rigorous sensitivity analysis on the decoding errors.
>     - **Experimental Setup:** We generated 1,000 watermarked images using stable diffusion v2.1. For each image, a 256-bit message was embedded, with each bit distributed across pseudo-randomly selected frequency bins.
>     - **Method:** We calculated the Off-Diagonal Correlation Matrix of the final bit decision errors across the 256 message bits over this dataset ($1,000 \times 256$ data points).
>     - **Result (Diffusion Process):** To verify independence between bit indices, we constructed a $256 \times 256$ Pearson correlation matrix of the bit error vectors. Specifically, for each bit index $i$, we aggregated the error outcomes across 1,000 images and computed its correlation with every other bit index $j$
>     - **Result (Under Attacks):** Furthermore, we stressed this assumption under spatially correlated distortions. Even under **J**PEG compression and Gaussian Blur, the average correlations remained low (**0.051** and **0.030**, respectively). This demonstrates that our distributed encoding maintains statistical independence not only against the diffusion prior but also against common frequency-dependent attacks.

---

> ### Author Response · Authors · 2025-11-22
>
> **Q2: The paper's empirical scope is limited to UNet-based Stable Diffusion models... The mid-band embedding is empirically justified, but its effectiveness may not generalize to different noise schedules or latent dimensions.**
>
> A2: We are currently finalizing the full-scale experiments on the expanded models to validate these findings comprehensively. We will post the complete quantitative results in the discussion thread immediately upon completion so that you can verify them before the final decision. Of course, these results will also be formally integrated into the final manuscript.
>
> We appreciate this valuable suggestion to expand our empirical scope. We agree that demonstrating generalization across architectures and sampling strategies is crucial. To address this, we conducted additional experiments on SDXL (different latent dimensions) and various schedulers (different noise schedules).
>
> 1. Generalization to Different Latent Dimensions: To verify robustness across latent dimensions, we applied PQIM to **Stable Diffusion XL (SDXL)**. SDXL operates on a latent space of $128 \times 128$, which is **4$\times$ larger** than the standard SD v2.1, v2.0, v1.4 ($64 \times 64$).
>     - Table (TPR@1%FPR/Bit Acc)
>
>     | **SDXL with 1024bits** | **No Distortion** | **JPEG (25)** | **Gaussian Blur (5)** | **Gaussian Noise (0.1)** | **Brightness (2)** | **Resize (0.5)** | **SP Noise (0.2)** | **Average** |
>     | --- | --- | --- | --- | --- | --- | --- | --- | --- |
>     | **PQIM** | 1.0000 / 1.0000 | 1.0000 / 0.9496 | 1.0000 / 0.8485 | 0.9980 / 0.7777 | 1.0000 / 0.9955 | 1.0000 / 0.9992 | 1.0000 / 0.7310 | 0.9997 / 0.9002 |
>     - The larger latent dimension provides a richer phase spectrum capacity, allowing PQIM to maintain robustness without modification. This confirms PQIM's scalability to high-resolution latent models.
> 2. **Robustness Across Noise Schedules:** To test sensitivity to the denoising trajectory, we evaluated PQIM on SD v2.1 using three distinct schedulers: PNDM, DDIM, and DPM-Solver.
>     - Table (TPR@1%FPR/Bit Acc)
>
>
>         | **SD2.1** | **No Distortion** | **JPEG (25)** | **Gaussian Blur (5)** | **Gaussian Noise (0.1)** | **Brightness (2)** | **Resize (0.5)** | **SP Noise (0.2)** | **Average** |
>         | --- | --- | --- | --- | --- | --- | --- | --- | --- |
>         | **PNDM** | 1.0000 / 1.0000 | 1.0000 / 0.9823 | 1.0000 / 0.9198 | 0.9770 / 0.8179 | 0.9990 / 0.9972 | 1.0000 / 0.9998 | 0.9970 / 0.8346 | 0.9955 / 0.9282 |
>         | **DDIM** | 1.0000 / 1.0000 | 1.0000 / 0.9823 | 1.0000 / 0.9197 | 0.9760 / 0.8178 | 0.9990 / 0.9972 | 1.0000 / 0.9998 | 0.9970 / 0.8346 | 0.9953 / 0.9282 |
>         | **DPMSolver** | 1.0000 / 1.0000 | 1.0000 / 0.9850 | 1.0000 / 0.9303 | 0.9070 / 0.8219 | 0.9980 / 0.9970 | 1.0000 / 0.9999 | 0.9640 / 0.8417 | 0.9784 / 0.9168 |
>     - All schedulers maintain strong robustness. This indicates that the phase-embedded signal is structurally resilient and persists regardless of the specific differential equation solver used for sampling.
>
> ---
>
> **Q3: Example of failure case?**
>
> A3: Thank you for this valuable suggestion. You are correct that phase modulation, if excessive, can alter structural semantics. We have addressed this by conducting a stress test to identify the perceptibility boundary and providing a failure case example. We intentionally pushed the payload capacity far beyond the standard operating limit to embed watermark bits into the majority of the latent space. We will include these visual examples in the final manuscript.

---

> ### Author Response · Authors · 2025-11-28
>
> We thank you for your thorough review. We have revised the text to address your concerns about fidelity metrics, model generalization and theoretical clarifications.
>
> 1. **Weakness 1: Quantitative Fidelity Metrics**
>     - **Response:** We have strengthened our fidelity evaluation as requested.
>         - We added **FID and CLIP scores** to **Table 1** to provide a quantitative comparison of generation quality.
>         - We also added **Table 3** to report extensive perceptual metrics for the perceptual tuning task.
> 2. **Weakness 3 & Question 1: Theoretical Assumptions and Hyperparameters**
>     - **Response:**
>         - **Distributed Encoding:** We revised **Section 4.2 ("Validation of distributed encoding")** to clarify the theoretical validation of our error bounds and address the i.i.d. assumption.
>         - **Quantization Step:** We expanded the explanation of the **"Impact of quantization step" in Section 4.2** to provide more detail on how this parameter balances robustness and fidelity.
> 3. **Question 2: Generalization (Samplers, Inversion, Architectures)**
>
> - **Response:** We have extensively expanded our experiments to demonstrate generalization:
>     - **Samplers & Inversion:** We added **Appendix A.2**, which includes "Compatibility with Various Sampling Methods" and an "Ablation on inference and inversion steps," showing PQIM's stability across different settings.
>     - **New Architectures:** We added **Appendix A.3**, featuring "Extension to DiT-based diffusion models" (PixArt-$\alpha$) and the "Impact of latent resolution on message scalability" (SDXL), confirming our method works on Transformer-based and larger models.
> 1. **Question 3 & Scalability: Failure Cases and Capacity**
>     - **Response:**
>         - **Scalability:** We extended our evaluation to include **1024-bit messages in Table 6**, demonstrating the limits and capabilities of our method.
>         - **Failure Analysis:** We added **Appendix A.7** to transparently discuss and visualize failure cases under extreme stress tests, as requested.

---

> ### Author Response · Authors · 2025-12-01
>
> **W2: Robustness testing omits geometric operations like rotation, cropping, or scaling. It is unclear if PQIM maintains superiority under these.**
>
> A2: We appreciate your suggestion to examine geometric robustness more closely. We have now included rotation and resize & crop and these results clarify two important points.
>
> 1. **Geometric attacks are a shared limitation of current semantic watermarkers.** We adopt the same familiy of distortions as prior works and then extend this protocol by explicitly evaluating resized and crop and rotation to probe geometric robustness. All compared schemes (generative watermarkers) experience severe degradation under moderate geometric misalignment. This result shows that PQIM is not uniquely fragile, rather, geometric misalignment remains a common open challenge for latent semantic watermarking.
>     - Robustness (TPR@1%FPR/Bit Accuracy)
>
>
>         | **Attack Type** | **Parameter** | **PQIM** | **TR** | **GS** | **PRC** |
>         | --- | --- | --- | --- | --- | --- |
>         | **Resized & Crop** | $0.75$ | 0.0110 / 0.5113 | 0.0060 / - | 0.0600 / 0.5242 | 0.0050 / 0.0045 |
>         | **Rotation** | $22.5^\circ$ | 0.0070 / 0.5043 | 0.0000 / - | 0.0110 / 0.5033 | 0.0070 / 0.0045 |
> 2. **Inherent trade-off: structure in phase vs. geometric invariance.** PQIM embeds bits in the latent phase spectrum, which controls the spatial structure of the generated image. As a consequence, it is sensitive to coordinate perturbations such as rotation and cropping. This is the unavoidable cost of our design choice to prioritize generative robustness and security.
>
> ---
>
> 1. **Security over built-in geometric invariance.** Achieving strong intrinsic geometric invariance typically requires embedding spatially synchronized, easily alignable patterns. However, such globally structured patterns create a predictable attack surface, making them easier to erase or spoof once their structure is known. In contrast, PQIM selects a pseudo-random, key- and nonce-dependent subspace in the frequency domain. This design favors cryptographic security and generative resilience over built-in geometric invariance.
> 2. **Not ragile to all geometricchanges.** Finally, we emphasize that PQIM is not brittle to all geometric operations. It achieves TPR@1%FPR=1.0000 under scaling (resize without cropping), showing that moderate resampling alone does not break our watermark. The main failure modes are operations that fundamentally destroy spatial slignment (rotation/crop), where all current semantic watermarkers degrade similarly.

---

### Official Review · Reviewer_m1bt · 2025-11-03

**Soundness:** 3
**Presentation:** 3
**Contribution:** 3
**Rating:** 6
**Confidence:** 4

**Summary:**

The proposed PQIM framework presents an interesting advancement in the field of semantic watermarking for diffusion models. Its combination of distributed encoding in phase spectrum, experimental validation, and security-focused design makes it a strong contribution. However, addressing the limitations related to payload scalability, key management and more valid mathematical proof would further enhance its practicality and broader adoption in real-world applications.

**Strengths:**

1. **Novel Approach**: The proposed Phase-Quantization Invisible Marking (PQIM) introduces an approach to semantic watermarking by embedding information in the phase spectrum of diffusion model latent noise. This strategy avoids the common pitfalls of previous methods, such as predictable frequency patterns in Tree-ring, enhancing robustness against targeted attacks.
2. **Security Focus**: By introducing a key-dependent secret subspace and avoiding uniform patterns, the proposed watermarking scheme eliminates vulnerabilities commonly found in existing methods. This makes the watermark more resilient to adversarial attacks that exploit structural predictability.
3. **Nice discovery:** The authors report that in the frequency domain of the latent space, phase governs texture and structural information, while magnitude defines energy distribution. This is an interesting finding, which provides a direction for future watermarking research to operate in the frequency domain for achieving fidelity.
4. **Enough Experiments**: The extensive set of experiments demonstrates that PQIM consistently outperforms existing watermarking techniques under various attacks, including regeneration and signal-processing distortions. The ablation studies, comparative robustness analysis, and perceptual evaluation all reinforce the method’s effectiveness in both maintaining image fidelity and robustness.
5. **Flexibility**: The scalability of PQIM to different payload sizes, as shown by its performance across a range of 8 to 512 bits, highlights its versatility. This is crucial for real-world applications where watermark capacity requirements may vary.

**Weaknesses:**

1. While the method shows robustness across various payload sizes from 8 to 512, the paper mentions that PQIM’s performance gradually degrades as the payload increases. Further exploration into optimizing the system for larger payloads without sacrificing robustness could enhance its practical applicability in data-rich real-world scenarios.
2. The claim that this is "the first semantic watermarking framework with provable guarantees on bit accuracy" is overstated. The mathematical proof for this only demonstrates that the more redundant backups, the lower the error rate. This shares the same motivation as other methods that back up watermark information like Gaussian Shading—namely, minimizing bit error rates through voting as much as possible. Therefore, I think this is not an innovation, but rather an application of engineering experience.
3. The method’s reliance on a secret key for the watermark embedding and extraction process, while providing security, may create practical challenges in key management, particularly in distributed systems. In the scenario of tracing the generated content, it is necessary to obtain the user's key in advance to extract the watermark information, requiring that you have already identified the user identity.

**Questions:**

see above

---

> ### Author Response · Authors · 2025-11-22
> **Response to Reviewer m1bt**
>
> We thank the reviewer for your constructive feedback and the time dedicated to reviewing our work. We are encouraged by your recognition of our work's **novel approach in the phase spectrum**, **security-focused design**, and **extensive experimental validation**.
>
> We acknowledge the valid concerns regarding payload scalability, the specific phrasing of our theoretical claims, and key management scenarios. In the following detailed responses, we clarify these points and demonstrate that **PQIM** remains a robust and practical solution for secure generative AI. We hope our response alleviates your concerns.
>
> ---
>
> **W1: While the method shows robustness,  PQIM’s performance gradually degrades as the payload increases.**
>
> A1: We appreciate this valuable observation. We acknowledge that performance degradation at higher payloads (beyond 512 bits) in Stable Diffusion v2.1 is primarily due to the spatial bandwidth constraints of its small latent resolution ($4 \times 64 \times 64$), rather than a fundamental limitation of the PQIM framework itself.
> To demonstrate that PQIM scales effectively with modern architectures, we conducted additional experiments comparing Stable Diffusion v2.1 with SDXL, which utilizes a larger latent space ($4 \times 128 \times 128$).
>
> 1. Impact of Latent Dimensions on Scalability (SD v2.1 vs SDXL): As hypothesized, the limitation is spatial. When we force higher payloads (1024 bits) into the small latent of SD v2.1, we must utilize higher frequency bands, which are naturally more vulnerable to distortions like blurring. However, SDXL provides $4\times$ the spatial bandwidth, allowing us to embed 1024 bits while maintaining high redundancy ($N _{bins}$) in robust frequency bands.
>
> - Table (TPR@1%FPR/Bit Acc)
>
>
>     | **Attack Type** | **Parameter** | **SD v2.1 (1024 bits)** | **SDXL (1024 bits)** | **Bit Acc. Gain** |
>     | --- | --- | --- | --- | --- |
>     | **No Distortion** | - | 1.0000 / 0.9397 | **1.0000 / 1.0000** | +6.02% |
>     | **JPEG** | $Q=25$ | 1.0000 / 0.8417 | **1.0000 / 0.9496** | +10.79% |
>     | **Gaussian Blur** | $r=5$ | 1.0000 / 0.7396 | **1.0000 / 0.8485** | +10.89% |
>     | **Gaussian Noise** | $std=0.1$ | 0.9760 / 0.6552 | **0.9980 / 0.7777** | +15.75% |
>     | **Brightness** | $\times 2.0$ | 0.9980 / 0.9065 | **1.0000 / 0.9955** | +8.90% |
>     | **Resize** | $0.5x$ | 1.0000 / 0.9165 | **1.0000 / 0.9992** | +8.27% |
>     | **SP Noise** | $prob=0.2$ | 1.0000 / 0.7095 | **1.0000 / 0.7310** | +2.94% |
> - As shown, SDXL achieves near-perfect accuracy (>99%) even with a 1024-bit payload under non-destructive conditions and maintains significantly higher robustness under attacks compared to SD v2.1. This confirms that PQIM’s capacity scales with the generative model's architecture. Furthermore, as discussed in our response to another reviewer, employing advanced Error Correction Codes and Soft-Decision decoding can further enhance the payload capacity within the same latent dimension by significantly improving spectral efficiency.

---

> ### Author Response · Authors · 2025-11-22
>
> **W2: The claim that this is "the first... with provable guarantees" is overstated. The mathematical proof shares the same motivation as Gaussian Shading. I think this is not an innovation, but rather an application of engineering experience.**
>
> A2: We appreciate your critique regarding the precision of our claims. We agree that the phrase "provable guarantees" could be interpreted too broadly. We will revise this to **"theoretically grounded error bounds"** in the final manuscript to accurately reflect our contribution.
>
> However, we would like to clarify that our work offers a distinct advancement beyond the general engineering experience of minimizing errors through voting. While we share the motivation of redundancy with existing methods (e.g., Gaussian Shading), the theoretical foundation differs significantly:
>
> 1. **Establishing a Predictive Framework (QIM Theory)**
> The core contribution of PQIM is not the use of redundancy (voting) itself, but the derivation of redundancy from Channel Coding Theory.
>     - **Existing Methods (Empirical Redundancy):** Most semantic watermarking methods employ redundancy implicitly or empirically to survive distortions. The required redundancy level is often determined through *post-hoc* experiments.
>     - **PQIM (Analytical Design):** We apply Quantization Index Modulation (QIM) to model the watermarking process as a communications channel. Theorem 1 establishes a direct mathematical relationship between the quantization step size ($\Delta$), redundancy ($N_{bins}$), and the Bit Error Rate (BER).
> 2. **Why This Matters (Predictability vs. Trial-and-Error)**
> This theoretical grounding transforms watermarking from a "trial-and-error" engineering task into a predictive design problem.
>     - Using our framework, a service provider can calculate the necessary hyperparameters ($\Delta, N_{bins}$) to satisfy a specific target BER given a noise model, prior to conducting extensive empirical sweeps.
>
> ---
>
> **W3: The method’s reliance on a secret key may create practical challenges requiring that you have already identified the user identity.**
>
> A3: Thank you for highlighting this practical concern regarding key management. We understand the concern that requiring a user-specific key for extraction seems to imply a circular dependency (needing the identity to get the key to find the identity).
>
> However, we would like to clarify that in practical deployment scenarios for generative AI services (e.g., API-based platforms), the burden of key management lies with the service provider, not the end-user. We envision two primary application scenarios that resolve this potential paradox:
>
> - **Scenario 1: Model Provenance (Copyright Protection)**
>     - **Objective:** To certify that an image was generated by a specific model (e.g., verifying "Is this image from Model X?").
>     - **Key Management:** The service provider uses a single global secret key for all generations from that model.
>     - **Verification:** The verifier (or the provider themselves) attempts extraction using this single known global key. If the watermark is detected, the model source is authenticated. No user identity or unique user key is required in this context.
> - **Scenario 2: User Tracing (Traitor Tracing)**
>     - **Objective:** To identify *which* specific user generated a leaked or misused image.
>     - **Key Management:** The Service Provider assigns a unique secret key ($K_{user}$) to each user.
>     - **Verification:** When a suspicious image is found, the service provider (who possesses the database of all user keys) performs the detection. The provider does not need to ask the user for the key. Instead, they iterate through the active user keys in their database. The key that successfully extracts a valid watermark signal identifies the specific user.
>     - **Feasibility:** While this requires computational effort proportional to the number of users, it effectively resolves the circular dependency. The identification is a search operation performed by the provider using their own records, ensuring that the user identity can be recovered solely from the image and the key database.

---

> > ### Comment · Reviewer_m1bt · 2025-11-27
> > **Thanks for the response**
> >
> > Thanks for the response and I will maintain my positive score.

---

> > > ### Author Response · Authors · 2025-11-28
> > >
> > > Dear Reviewer m1bt,
> > >
> > > We sincerely thank you for your response and for maintaining your positive assessment of our work.
> > >
> > > We have updated the revised manuscript to reflect your valuable feedback, specifically:
> > >
> > > 1. **Theoretical Claims:** We refined the language regarding "provable guarantees" in the Introduction to be more precise.
> > > 2. **Scalability:** We added an analysis of the impact of latent resolution on message scalability in **Appendix A.3**.
> > >
> > > We appreciate your time and constructive suggestions which have helped improve our paper.
> > >
> > > Best regards,
> > >
> > > Authors

---

### Author Response · Authors · 2025-11-28

Dear Reviewers,

We sincerely thank you for reviewing our paper and for your insightful comments and  valuable feedbak. We are encouraged by the positive comments regarding the novelty of our Phase-Quantization Invisible Marking (PQIM) and its security-focused design

- Novelty & Security: Training-free, model-agnostic framework with a key-dependenet secret subspace (Reviewers m1bt, UeoB, gnUq, p1ZU).
- Theoretical Depth: The use of phase spectrum governing texture/structure and the theoretical error bounds (Reviewers m1bt, UeoB, p1ZU).
- Effiectiveness: Consistent performance across various attacks and scalability in payload (Reviewers m1bt, gnUq).
- High Fidelity: Superior perceptual quality, described as “clearly outputting the perceptually closest result to the original” compared to baselines (Reviewers p1ZU).

We have actively revised the manuscript to address your concerns. The major updates include:

- Extended Experiments & Scalability (Response to Reviewers UeoB, m1bt).
- Methodological Clarifications & Theory (Response to Reviewers m1bt, p1ZU, UeoB).
- Presentation & Structure Improvements (Response to Reviewer p1ZU).

---

### Meta-Review · Area_Chair_iXWU · 2026-01-06

**Summary:**

The paper proposes a diffusion watermarking method based on phase modulation and demonstrates strong empirical robustness across a range of attacks.

However, reviewers express concern that the theoretical analysis is limited, relying on simplifying assumptions (e.g., independent errors) and falling short of rigorously explaining robustness to regeneration or adaptive removal attacks.

Given the mixed reviews and my own reading, the analysis does not establish a provable robustness guarantee. In particular, Theorem 1 primarily demonstrates exponential error decay under the independence assumption, while the proposed algorithm largely amounts to applying QIM in the phase space. As a result, the technical novelty appears limited, which motivates a recommendation for rejection.

**Reviewer Concerns:**

The rebuttal clarified implementation details, experimental protocols, and scope limitations, which addressed several presentation and evaluation concerns. However, the key issues regarding limited novelty and the depth of the theoretical contribution remain unresolved.

**Reviewer Scores:**

The scores are mixed. Reviewers who were initially positive would likely maintain their scores but did not express increased confidence after the discussion. The more critical reviewer(s) did not indicate any upward revision, and their concerns largely remain. Overall, the score distribution is expected to stay mixed.

---

### Decision · Program_Chairs · 2026-01-26

Reject